# Learning the Expected Core of
# Strictly Convex Stochastic Cooperative Games

**Nam Phuong Tran**
Department of Computer Science
University of Warwick
Coventry, United Kingdom
`nam.p.tran@warwick.ac.uk`

**The Anh Ta**
CSIRO's Data61
Marsfield, NSW, Australia
`theanh.ta@csiro.au`

**Shuqing Shi**
Department of Informatics
King's College London
London, United Kingdom
`shuqing.shi@kcl.ac.uk`

**Debmalya Mandal**
Department of Computer Science
University of Warwick
Coventry, United Kingdom
`debmalya.mandal@warwick.ac.uk`

**Yali Du**
Department of Informatics
King's College London
London, United Kingdom
`yali.du@kcl.ac.uk`

**Long Tran-Thanh**
Department of Computer Science
University of Warwick
Coventry, United Kingdom
`long.tran-thanh@warwick.ac.uk`

## Abstract

Reward allocation, also known as the credit assignment problem, has been an important topic in economics, engineering, and machine learning. An important concept in reward allocation is the core, which is the set of stable allocations where no agent has the motivation to deviate from the grand coalition. In previous works, computing the core requires either knowledge of the reward function in deterministic games or the reward distribution in stochastic games. However, this is unrealistic, as the reward function or distribution is often only partially known and may be subject to uncertainty. In this paper, we consider the core learning problem in stochastic cooperative games, where the reward distribution is unknown. Our goal is to learn the expected core, that is, the set of allocations that are stable in expectation, given an oracle that returns a stochastic reward for an enquired coalition each round. Within the class of strictly convex games, we present an algorithm named `Common-Points-Picking` that returns a point in the expected core given a polynomial number of samples, with high probability. To analyse the algorithm, we develop a new extension of the separation hyperplane theorem for multiple convex sets.

## 1 Introduction

The reward allocation problem is a fundamental challenge in cooperative games that seeks reward allocation schemes to motivate agents to collaborate or satisfy certain constraints, and its solution concepts have recently gained popularity within the machine learning literature through its application in explainable AI (XAI) [17, 28, 13, 31] and cooperative Multi-Agent Reinforcement Learning (MARL) [29, 12, 30]. In the realm of XAI, designers often seek to understand which factors of the model contribute to the outputs. Solution concepts such as the Shapley value [17, 28] and the core [31, 13] provide frameworks for assessing the influence that each factor has on the model's

38th Conference on Neural Information Processing Systems (NeurIPS 2024).

behavior. In cooperative MARL, these solution concepts offer a framework for distributing team rewards to individual agents, promoting cooperation among players, and preventing the occurrence of lazy learner phenomena [29, 12, 30]. A crucial notion of reward allocation is stability, defined as an allocation scheme wherein no agent has the motivation to deviate from the grand coalition. The set of stable allocations is called *the core of the game*.

In the classical setting, the reward function is deterministic and commonly known among all agents, with no uncertainty within the game. However, assuming perfect knowledge of the game is often unrealistic, as the outcome of the game may contain uncertainty. This led to the study of stochastic cooperative games, dated back to the seminal works of [9, 27], where stability can be satisfied either with high probability, known as the robust core, or in expectation, known as the expected core. However, in these works, the distribution of stochastic rewards is given, allowing agents to calculate the reward before the game starts, which is not practical since the knowledge of the reward distribution may only be partially known to the players. When the distribution of the stochastic reward is unknown, the task of learning the stochastic core by sequentially interacting with the environment appears much more challenging. Early attempts [19, 20] studied robust core and expected core learning under full-information, where full information means that the random rewards of all coalitions are observed at each round. However, full-information feedback is a strong assumption, which does not hold in many real-world situations. Instead, agents typically observe the reward of their own coalition only (e.g., only know the value of their own action - joining a particular coalition in this case), which is typically known as bandit feedback in the online learning literature.

In our work, we focus on learning the expected core, which circumvents the potential emptiness of the robust core in many practical cases. Moreover, instead of full-information feedback, where the stochastic rewards of all coalitions are observed each round, we consider the bandit feedback setting, where only the stochastic reward of the inquired coalition is observed each round. Given the lack of knowledge about the probability distribution of the reward function, learning the expected core using data-driven approaches with bandit feedback is a challenging task.

Against this background, the contribution of this paper is three-fold: **(1)** We focus on expected core learning problem with unknown reward function, and propose a novel algorithm called the `Common-Points-Picking` algorithm, the first of its kind that is designed to learn the expected core with high probability. Notably, this algorithm is capable of returning a point in an unknown simplex, given access to the stochastic positions of the vertices, which can also be used in other domains, such as convex geometry. **(2)** We establish an analysis for finite sample performance of the `Common-Points-Picking` algorithm. The key component of the analysis revolves around a novel extension of the celebrated hyperplane separation theorem, accompanied by further results in convex geometry, which can also be of independent interest. **(3)** We show that our algorithm returns a point in expected core with at least $1 - \delta$ probability, using $\mathrm{poly}(n, \log(\delta^{-1}))$ number of samples.

## 2 Related Work

**Stochastic Cooperative Games.**   The study of stochastic cooperative games can be traced back to at least [9]. The main goal of the allocation scheme is to minimise the probability of objections arising after the realisation of the rewards. [27, 26] later extended and refined the notion of stability in stochastic settings. These seminal works require information about the reward distribution to compute a stable allocation scheme before the game starts. Stochastic cooperative games have also been studied in a Bayesian setting in a series of papers [5, 7, 8, 6]. These works develop a framework for Bayesian stochastic cooperative games, where the distribution of the reward is conditioned on a hidden parameter following a prior distribution. The prior distribution is common knowledge among agents and can be used to define a new concept of stability called Bayesian core. In contrast to previous works, our paper focuses on studying scenarios where the reward distribution or prior knowledge is not disclosed to the principal agent and computing a stable allocation requires a data-driven method.

**Learning the Core.**   The literature on learning the core through sample-based methods can be categorised based on the type of core one seeks to evaluate. Two main concepts of the stochastic core are commonly considered, namely the robust core (i.e. core constraints are satisfied with high probability) [11, 22, 19] and the expected core (i.e. core constraints are satisfied in expectation) [10, 20]. The robust core is defined in a manner that allows the core inequalities to be satisfied with high probability when the reward is realised. However, this definition may lead to the practical issue

of an empty robust core, as illustrated in [10]. To mitigate the potential emptiness of the robust core, we investigate the learnability of the expected core.

The work most closely related to ours is [20], in which the authors introduce an algorithm designed to approximate the expected core using a robust optimization framework. In the context of full information feedback, where rewards for all allocations are revealed each round, the algorithm demonstrates asymptotic convergence to the expected core. Furthermore, the authors provide an finite-samples error bound for the distance to the expected core. However, when dealing with bandit feedback, where the reward of the enquired coalition is returned each round, naively applying the algorithm of [20] may result in an exponential number of samples in terms of the number of players. In the bandit feedback setting, as *we later establish in Theorem 7 that, in general cooperative games, no algorithm can learn the expected core with a finite number of samples without additional assumptions.* This highlights the key difference in dealing with bandit feedback compared to full-information feedback. Given this limitation, we narrow our focus to (strictly) convex games, an important class of cooperative games where the expected reward function is (strictly) supermodular. By leveraging strict convexity, we introduce the `Common-Points-Picking` algorithm, which efficiently returns a point in the expected core with high probability using only a polynomial number of samples. While [20] proposed a general algorithm applicable to strictly convex games, we argue that it lacks statistical and computational efficiency due to the absence of a mechanism to exploit the supermodular structure of the expected reward function. Further detailed comparison can be found in Appendix E.1.

**Learning the Shapley Value.** Another relevant line of research is the Shapley approximation, as the Shapley value is the barycenter of the core in convex games. Therefore, the approximation error of the Shapley value is an upper bound on the distance between the approximated Shapley value and the expected core [4, 18, 24]. It is worth noting that, in contrast to the Shapley approximation method, our algorithm ensures the return of a point in the core. In comparison, the Shapley approximation approach can only provide an allocation with a bounded distance from the core.

## 3 Problem Description

### 3.1 Preliminaries

**Notations.** For $k \in \mathbb{N}^+$, denote $[k]$ as set $\{1, 2, \ldots, k\}$. For $n \in \mathbb{N}^+$, let $\mathbf{E}^n$ be the $n$-dimensional Euclidean space, and let us denote $\mathcal{D}$ as the Euclidean distance in $\mathbf{E}^n$. Denote $\mathbf{1}_n$ as the vector $[1, ..., 1] \in \mathbb{R}^n$. Denote $\langle \cdot, \cdot \rangle$ as the dot product. For a set $C$, we denote $C \setminus x$ as the set resulting from eliminating an element $x$ in $C$. For $C \subset \mathbf{E}^n$, let $\text{diam}(C) := \max_{x,y \in C} \mathcal{D}(x, y)$, and $\text{Conv}(C)$ denote the diameter and the convex hull of $C$, respectively.

Denote $\mathfrak{S}_n := \{\omega : [n] \to [n] \mid \omega \text{ is a bijection}\}$ as the permutation group of $[n]$. For any collection of permutations $\mathcal{P} \subset \mathfrak{S}_n$, we denote $\omega_p$, $p \in [|\mathcal{P}|]$, as $p^{\text{th}}$ permutation order in $\mathcal{P}$. Let $s_i := (i, i+1)$ denote the *adjacent transposition* between $i$ and $i+1$. Given a set $C$, we denote by $\mathcal{M}(C)$ the space of all probability distributions on $C$.

**Stochastic Cooperative Games.** A *stochastic* cooperative game is defined as a tuple $(N, \mathbb{P})$, where $N$ is a set containing all agents with number of agents to be $|N| = n$, and $\mathbb{P} = \{\mathbb{P}_S \in \mathcal{M}([0, 1]) \mid S \subseteq N\}$ is the collection of reward distributions with $\mathbb{P}_S$ to be the reward distribution w.r.t. the coalition $S$. For any coalition $S \subseteq N$, we denote $\mu(S) := \mathbb{E}_{r \sim \mathbb{P}_S}[r]$ as the expected reward of coalition $S$. For a reward allocation scheme $x \in \mathbb{R}^n$, let $x(S) := \sum_{i \in S} x_i$ as the total reward allocation for players in $S$. A reward allocation $x$ is *effective* if $x(N) = \mu(N)$. The hyperplane of all effective reward allocations, denoted by $H_N$, is defined as $H_N = \{x \in \mathbb{R}^n \mid x(N) = \mu(N)\}$. The convex stochastic cooperative game can be defined as follows:

**Definition 1** (**Convex stochastic cooperative game**). A stochastic cooperative game is convex if the expected reward function is supermodular [23], that is,

$$\mu(S \cup \{i\}) - \mu(S) \geq \mu(C \cup \{i\}) - \mu(C)), \quad \forall i \notin S \cup C \text{ and } \forall C \subseteq S \subseteq N. \tag{1}$$

Next, we define the notion of strict convexity as follows:

**Definition 2** ($\varsigma$-**Strictly convex cooperative game**). For some constant $\varsigma > 0$, a cooperative game is $\varsigma$-strictly convex if the expected reward function is $\varsigma$-strictly supermodular [23], that is, $\mu$ is supermodular and

$$\mu(S \cup \{i\}) - \mu(S) \geq \mu(C \cup \{i\}) - \mu(C) + \varsigma, \quad \forall i \notin S \cup C \text{ and } \forall C \subseteq S \subseteq N. \tag{2}$$

Following [20], we define the expected core as follows:

**Definition 3 (Expected core [20]).** The core is defined as

$$\text{E-Core} := \{x \in \mathbb{R}^n \mid x(N) = \mu(N); \ x(S) \geq \mu(S), \ \forall S \subseteq N\}.$$

That is, the E-Core is the collection of all effective reward allocation schemes $x$ (i.e., schemes where the total allocation adds up to the expected reward of the grand coalition $N$ - see the definition of effective reward allocation above), where if any agents deviate from the grand coalition $N$ to form a smaller team $S$, regardless of how they allocate the reward, each individual will not receive more expected reward than if they had stayed in $N$. Note that, as E-Core $\subset H_N$, its dimension is at most $(n-1)$. We say that E-Core is *full dimensional* whenever its dimension is $n-1$.

In convex games, each vertex of the core in the convex game is a marginal vector corresponding to a permutation order [23]. This is a special property of convex games, which plays a crucial role in our algorithm design. For any $\omega \in \mathfrak{S}_n$, define the marginal vector $\phi^\omega \in \mathbb{R}^n$ corresponding to $\omega$, that is, its $i^{\text{th}}$ entry is

$$\phi_i^\omega := \mu(P^\omega(i)) - \mu(P^\omega(i) \setminus i), \tag{3}$$

where $P_i^\omega = \{j \mid \omega(j) \leq \omega(i)\}$. It is known from [23] that if the expected reward function is strictly supermodular, the *E-Core must be full dimensional*.

## 3.2 Problem Setting

In our setting we assume that there is a principal who does not know the reward distribution $\mathbb{P}$. In each round $t$, the principal queries a coalition $S_t \subset N$. The environment returns a vector $r_t \sim \mathbb{P}_{S_t}$ independently of the past. For simplicity, we assume that the agent knows the expected reward of the grand coalition $\mu(N)$. Additionally, we assume that the convexity of the game, that is, $\mu$ is supermodular. Our question is how many samples are needed so that with high probability $1 - \delta$, the algorithm returns a point $x \in$ E-Core. More particularly, we ask whether or not there is an algorithm that can output a point in the E-Core, with probability at least $1 - \delta$ and the number of queries

$$T = \text{poly}(n, \log(\delta^{-1})). \tag{4}$$

As well shall show in Theorem 7, if E-Core is not full-dimensional, no algorithm can output a point in E-Core with finite samples. As such, to guarantee the learnability of the E-Core. From now on in the rest of this paper, we assume that:

**Assumption 4.** *The game is $\varsigma$-strictly convex.*

Note that *strict* convexity immediately implies full dimensionality [23], which is not the case with convexity (refer to Section 5). As we shall show in the next sections, strict convexity is a sufficient condition allowing the principal to learn a point in E-Core with polynomial number of samples. Practical examples of strictly convex games can be found in appendix D.

# 4 Learning the Expected Core

In this section, we propose a general-purpose `Common-Point-Picking` algorithm that is able to return a point of unknown full-dimensional simplex, given an oracle that provides a noisy position of the simplex's verticies. Under the assumption that the game is strictly convex, we show that, when applying `Common-Points-Picking` algorithm to the $\varsigma$-strictly convex game, it can return a point in E-Core provided the number of samples is $\text{poly}(n, \varsigma)$.

## 4.1 Geometric Intuition

Given E-Core polytope of dimension $(n-1)$. In deterministic case when the knowledge of the game is perfect, to compute a point in the core, one can query a marginal vector corresponding to a permutation order $\omega \in \mathfrak{S}_n$ [23]. Given that we have uncertainty in the estimation of E-Core, this approach is no longer applicable. The reason is that for each vertex $\phi^\omega$, we do not precisely compute its position. Instead, we only have information on its confidence set $\mathcal{C}(\phi^\omega, \delta)$, a compact $(n-1)$-dimensional set. The confidence set contains $\phi^\omega$ with probability at least $(1 - \delta)$, as we will define shortly in this section.

One approach to overcome the effect of uncertainty is that we can estimate multiple vertices of the E-Core. Let $\mathcal{P} \subset \mathfrak{S}_n$ be a collection of permutations, and $Q = \{\phi^{\omega_p} \mid \omega_p \in \mathcal{P}\}$ be the set of

vertices corresponding to $\mathcal{P}$. For brevity, we denote $\mathcal{C}_p := \mathcal{C}(\phi^{\omega_p}, \delta)$, $\omega_p \in \mathcal{P}$. In this section, we assume that the confidence sets contain the true vertices, that is, $\phi^{\omega_p} \in \mathcal{C}_p$, $\forall p \in [|\mathcal{P}|]$, which can be guaranteed with high probability. It is clear that $\mathrm{Conv}(Q) \subset$ E-Core, since $Q$ is a subset of vertices of E-Core. The challenge is that, as the ground truth of the position of the vertex can be any point in the confidence set, we need to ensure that the algorithm outputs a point in the convex hull of any collection of points in the confidence sets. A sufficient condition to achieve this is that, given $|\mathcal{P}|$ confidence sets $\{\mathcal{C}_p\}_{p \in [|\mathcal{P}|]}$, for each $x^p \in \mathcal{C}_p$,

$$\bigcap_{\substack{x^p \in \mathcal{C}_p \\ p \in [|\mathcal{P}|]}} \mathrm{Conv}\left(\{x^p\}_{p \in [|\mathcal{P}|]}\right) \neq \varnothing. \tag{5}$$

This condition means that there exists a common point among all the convex hulls formed by choosing any point in confidence sets, $x^p \in \mathcal{C}_p$. We call the above intersection a set of common points. The reason why it is a sufficient condition is that this set is a subset of a ground-truth simplex, implying that it is in the E-Core. Formally, we have

$$\bigcap_{\substack{x^p \in \mathcal{C}_p \\ p \in [|\mathcal{P}|]}} \mathrm{Conv}\left(\{x^p\}_{p \in [|\mathcal{P}|]}\right) \subset \mathrm{Conv}(Q) \subset \text{E-Core}; \tag{6}$$

which means that any common point must be in E-Core. Moreover, the set of common points *is learnable*. As we shall show in the next section, our algorithm can access the set of common points whenever it is nonempty.

We first state a necessary condition for the number of vertices of E-Core need to estimate for (5) can be satisfied:

**Proposition 5.** *Assume that all the confidence sets are full dimensional, that is,* $\dim(\mathcal{C}_p) = n - 1$, $\forall p \in [|\mathcal{P}|]$, *and suppose that* $|\mathcal{P}| < n$,

$$\bigcap_{\substack{x^p \in \mathcal{C}_p \\ p \in [|\mathcal{P}|]}} \mathrm{Conv}\left(\{x^p\}_{p \in [|\mathcal{P}|]}\right) = \varnothing. \tag{7}$$

Proposition 5 implies that one needs to estimate at least $n$ vertices to guarantee the existence of a common point.

---

**Algorithm 1** Common Points Picking

1: Input collection of permutation order $\mathcal{P} = \{\omega_p\}_{p \in [n]}$.
2: $t = 0$, ep $= 0$, $Q = \varnothing$
3: **while** `Stopping-Condition` $(Q, b_{\mathrm{ep}})$ **do**
4:    ep $\leftarrow$ ep $+ 1$;
5:    **for** $p \in [n]$ **do**
6:       **for** $i \in [n]$ **do**
7:          Query $P_i^{\omega_p}$.
8:          Orcale returns $r_{\mathrm{ep}}\left(P_i^{\omega_p}\right) \leftarrow r_t$.
9:          $t \leftarrow t + 1$.
10:         Computing $\hat{\phi}_i^{\omega_p}(\mathrm{ep})$ as (9).
11:       **end for**
12:    **end for**
13:    Assign $Q = \left\{\hat{\phi}^{\omega_p}(\mathrm{ep})\right\}_{p \in [n]}$
14:    Compute $b_{\mathrm{ep}}$ as (10)
15: **end while**
16: Return $x^\star = \frac{1}{n} \sum_{\omega \in \mathcal{P}} \hat{\phi}^\omega(\mathrm{ep})$.

---

### 4.2 `Common-Points-Picking` **Algorithm**

As Proposition 5 suggests, we need to estimate at least $n$ vertices. As such, from now on, we assume that $|\mathcal{P}| = n$. Based on the above intuition, we propose `Common-Points-Picking`, whose pseudo code is described in Algorithm 1.

The `Common-Points-Picking` Algorithm can be described as follows. First, let us explain the notation. In epoch ep, the variable $r_{\mathrm{ep}}(\cdot)$ in the algorithm represents the reward value of the enquired coalition; $\hat{\phi}^{\omega_p}(\mathrm{ep})$ represents the estimated marginal vector w.r.t. $\omega_p$. In each epoch ep, assuming that the stopping condition is not satisfied, the algorithm estimates the marginal vectors corresponding to the collection of given permutation orders $Q = \left\{\hat{\phi}^{\omega_p}(\mathrm{ep})\right\}_{p \in [n]}$ (lines 6-10). For each $p \in [n]$, the estimation can be done by querying the value of the nested sequence $P_1^{\omega_p}$, $P_2^{\omega_p}$, ..., $P_n^{\omega_p}$ (line 6-7), then estimating the marginal contribution of each player with respect to the permutation order $\omega_p$ (line 10). Next, it calculates the confidence bonus $b_{\mathrm{ep}}$ of the confidence sets and checks the stopping condition for the next epoch. The algorithm continues until the stopping condition is satisfied, and then returns the average of the most recent values of the marginal vectors corresponding to $\mathcal{P}$.

The termination of the `Common-Points-Picking` algorithm is based on the `Stopping-Condition` algorithm (Algorithm 2), which can be described as follows. Consider the case where $Q \neq \varnothing$. For each point $x^p \in Q$, the algorithm attempts to calculate the separating hyperplane $H_p(Q)$, that separates $x^p$ from the rest $Q \setminus x^p$ (line 7).

The hyperplane $H_p(Q)$ is defined by two parameters $(v^p \in \mathbb{R}^n, c^p \in \mathbb{R})$, where $v^p$ is one of its unit normal vectors, together with $c^p$, satisfying Eq. (8). Specifically, the second and third equality in (8) implies that $H_p(Q)$ is parallel and at a distance of $\epsilon_{\text{ep}}$ (toward $x^p$) from the hyperplane that passes through all the points in $Q \setminus x^p$. The fourth equality in (8) guarantees that $H_p(Q)$ is a subset of $H_N$ (as the normal vector of $H_N$ is $\mathbf{1}_n$). After computing $H_p(Q)$, the algorithm checks whether the distance from the confidence set $\mathcal{C}_p$ to $H_p(Q)$ is large enough (lines 8-12). The stopping condition checks for all $p \in [n]$; if no condition is violated, then the algorithm returns TRUE. An example of the construction of separating hyperplanes in $H_N$, where $n = 3$ is depicted in Figure 1.

Note that the input of algorithm $\mathcal{P}$ can be any collection of permutation orders such that $|\mathcal{P}| = n$. In the next section, we will provide instances of the collection of permutation orders, in which, under Assumption 4, the algorithm can output a point in E-Core with high probability and a polynomial number of samples.

**Remark 6.** Most of the computational burden lies in computing the separating hyperplane $H_p(Q)$ for each $p$ (line 7), and calculating the distance between the confidence set $\mathcal{C}_p$ and $H_p(Q)$ (line 8) in Stopping Condition. Since all tasks can be completed within polynomial time w.r.t. $n$, our algorithm is polynomial.

---

**Algorithm 2** Stopping Condition

1: Input collection of $n$ points $Q$, confidence bonus $b_{\text{ep}}$.
2: Compute $\epsilon_{\text{ep}} = 2\sqrt{n}b_{\text{ep}}$.
3: **if** $Q = \varnothing$ **then**
4:    Return FALSE.
5: **end if**
6: **for** $p \in [n]$ **do**
7:    Computing $H_p(Q)$, i.e., compute $(v^p \in \mathbb{R}^n, c^p \in \mathbb{R})$ such that.

$$\begin{cases} \|v^p\|_2 & = 1; \\ \langle v^p, x \rangle & = c^p + \epsilon_{\text{ep}}, \forall x \in Q \setminus x^p. \\ \langle v^p, x^p \rangle & < c^p + \epsilon_{\text{ep}}; \\ \langle v^p, \mathbf{1}_n \rangle & = 0; \end{cases} \quad (8)$$

8:    Computing distance:

$$h_p := \min_{x : \|x - x^p\|_\infty < b_{\text{ep}}} c^p - \langle v^p, x \rangle.$$

9:    **if** $h_p < n \, \epsilon_{\text{ep}}$ **then**
10:        Return FALSE.
11:    **end if**
12: **end for**
13: Return TRUE.

---

There are two challenges regarding the Common-Points-Picking algorithm. First, we need to design confidence sets such that they contain the vertices with high probability, which can be done easily using Hoeffding's inequality. Second, we need to define a stopping condition that can guarantee the non-emptiness of the common set and output a point in the common set with a polynomial number of samples. The second question is *more involved and requires proving new results in convex geometry*, including an extension of the hyperplane separation theorem, as we shall fully explain in Section 5.1.

**Confidence Set** To calculate this set we will use the probability concentration inequality to obtain a confidence set: First, let $r_{\text{ep}}(\varnothing) = 0, \forall \text{ep} > 0$, define the empirical marginal vector w.r.t. permutation $\omega$ as $\hat{\phi}^\omega \in \mathbb{R}^n$ at epoch ep as

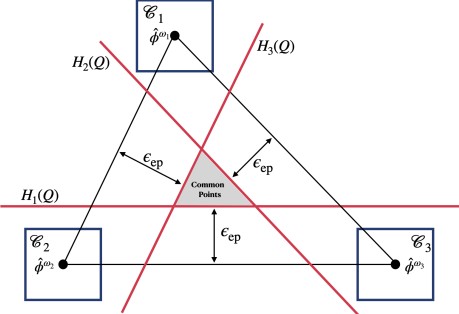

Figure 1: Set of common points constructed by separating hyperplanes. The intersection of half-spaces defined by $H_p(Q)$, $\forall p \in [n]$, creates a subset of common points. The common points are in E-Core, provided that the confidence sets contain the ground-truth vertices.

$$\hat{\phi}_i^\omega(\text{ep}) = \frac{1}{\text{ep}} \sum_{s=1}^{\text{ep}} r_s \left( P_i^\omega \right) - r_s \left( P_i^\omega \setminus i \right). \quad (9)$$

By the Hoeffding's inequality, one has that after ep epochs, $\forall \omega \in \mathcal{P}$, for each $i \in [n]$, with probability at least $1 - \delta$, $\phi^\omega$ must belong to the following set:

$$\mathcal{C}(\phi^\omega, \delta) = \left\{ x \in H_N \,\middle|\, \left\| x - \hat{\phi}^\omega \right\|_\infty \leq b_{\text{ep}} \right\}; \quad \text{s.t.} \quad b_{\text{ep}} := \sqrt{\frac{2\log(n \, \text{ep} \, \delta^{-1})}{\text{ep}}}. \quad (10)$$

# 5 Main Results

Before proceeding to the analysis of Algorithm 1, let us exclude the case where learning a stable allocation is not possible, thereby emphasizing the need of the strict convexity assumption.

**Theorem 7.** *Suppose that E-Core has dimension $k < n - 1$, for any $0.2 > \delta > 0$ and with finite samples, no algorithm can output a point in E-Core with probability at least $1 - \delta$.*

The proof of Theorem 7 employs an information-theoretic argument. In particular, we construct two game instances with low-dimensional cores, utilising the concept of face games introduced in [14]. The construction of the two games is designed to ensure that, despite the KL distance between their reward distributions being arbitrarily small, their low-dimensional cores are parallel and maintain a positive distance from each other. Consequently, their cores have empty intersections. This implies that, given finite samples, no algorithm can reliably distinguish between the two games. Since these games lack a mutually stable allocation, if an algorithm fails to differentiate between them, it is likely to choose the wrong core with a certain probability.

It is worth noting that convex games may have a low-dimensional core, as demonstrated in the following example.

**Example 8.** Let $\mu(S) = |S|$ for all $S \subseteq N$. It is easy to verify that $\mu$ is indeed convex. The marginal contribution of any player $i$ to any set $S \subseteq N$ is

$$\mu(S \cup i) - \mu(S) = 1, \ \forall S \subset N. \tag{11}$$

Therefore, the only stable allocation is $\mathbf{1}_n$, which coincides with the Shapley value. Hence, the core is one-point set. According to Theorem 7, since the core has a dimension of $0$ in this case, it is impossible to learn a stable allocation with a finite number of samples.

Example 8 suggests that convexity alone does not ensure the problem's learnability, emphasizing the requirement for strict convexity.

## 5.1 On the Stopping Condition

In this subsection, we explain the construction of the stopping condition in Algorithm 2. We will show that the stopping condition can always be satisfied with the number of samples needed polynomially w.r.t the width of the ground truth simplex. Intuitively, the confidence sets need to shrink to be sufficiently small, relative to the width of the simplex, to guarantee the existence of a common point.

To simplify the presentation, we restrict our attention to $H_N$ and consider it as $\mathbf{E}^{n-1}$. First, we state a necessary condition for the existence of common points.

**Proposition 9.** *Suppose there is a $(n-2)$-dimensional hyperplane that intersect with all the interior of confidence sets $\mathcal{C}_p, \ \forall p \in [n]$, then common points do not exist.*

Proposition 9 suggests that if the ground truth simplex $\mathrm{Conv}\,(Q)$ is not full-dimensional, then the common set is empty. In addition, even if $\mathrm{Conv}\,(Q)$ is full-dimensional, but the confident sets are not sufficiently small, one can also create a hyperplane that intersects with all the confidence sets. For example, when the intersection of the confidence sets is not empty.

On the other hand, when the confidence sets are well-arranged and sufficiently small, that is, there does not exist a hyperplane that intersects with all of them, a nice separating property emerges, as stated in the next theorem. This *new result can be considered as an extension of the classic separating hyperplane theorem* [3]. First, let us recap the notion of separation as follows.

**Definition 10 (Separating hyperplane).** Let $C$ and $D$ be two compact and convex subsets of $\mathbf{E}^{n-1}$. Let $H$ be a hyperplane defined by the tuple $(v, c)$, where $v$ is a unit normal vector and $c$ is a real number, such that $\langle x, v \rangle = c, \ \forall x \in H$. We say $H$ separates $C$ and $D$ if $\langle x, v \rangle > c, \ \forall x \in C$; and $\langle y, v \rangle < c, \ \forall y \in D$.

**Theorem 11 (Hyperplane separation theorem for multiple convex sets).** *Assume that $\{\mathcal{C}_p\}_{p \in [n]}$ are mutually disjoint compact and convex subsets in $\mathbf{E}^{n-1}$. Suppose that there does not exist a $(n-2)$-dimensional hyperplane that intersects with confidence sets $\mathcal{C}_p, \ \forall p \in [n]$, then for each $p \in [n]$, there exists a hyperplane that separates $\mathcal{C}_p$ from $\bigcup_{q \neq p} \mathcal{C}_q$.*

**Remark 12 (Non-triviality of Theorem 11).** At a first glance, Theorem 11 may appear as a trivial extension of the classic hyperplane separation theorem due to the following reasoning: Consider the union of all hyperplanes that intersect $\bigcup_{q\neq p} \mathcal{C}_q$, which trivially contains $\bigcup_{q\neq p} \mathcal{C}_q$. Then, by assuming that these hyperplanes do not intersect $\mathcal{C}_p$, the separation between $\mathcal{C}_p$ and $\bigcup_{q\neq p} \mathcal{C}_q$ appears to follow from the classic separation hyperplane theorem. However, there is a flaw in the above reasoning: The union of these hyperplanes is *not necessarily convex*. Therefore, the classic separation hyperplane theorem cannot be applied directly. Instead, employing Carathéodory's theorem, we prove in Theorem 11 by contra-position that if the intersection between $\mathcal{C}_p$ and $\text{Conv}(\bigcup_{q\neq p} \mathcal{C}_q)$ is non-empty, then we can construct a low-dimensional hyperplane that intersects with all the set.

When those confidence sets are well-separated, we can provide a sufficient condition for that the common points exist. Let $H_p$ be a separating hyperplane that separate $\mathcal{C}_p$ from $\bigcup_{q\neq p} \mathcal{C}_q$. We define $H_p$ corresponding with tuple $(v^p, c^p)$. Now, denote $E_p = \{x \in \mathbf{E}^{n-1} \mid \langle v^p, x \rangle < c^p\}$ as the half space containing $\mathcal{C}_p$. We have that:

**Lemma 13.** *For any $x^p \in \mathcal{C}_p$, $p \in [n]$,*

$$\bigcap_{p\in[n]} E_p \subseteq \text{Conv}\left(\{x^p\}_{p\in[n]}\right). \tag{12}$$

*Consequently, if $\bigcap_{p\in[n]} E_p$ is nonempty, it is the subset of common points.*

From Lemma 13, as $\bigcap_{p\in[n]} E_p$ is the subset of any simplex defined by a set of points in the confidence sets, $\bigcap_{p\in[n]} E_p$ must be either empty or bounded subset of $\mathbf{E}^{n-1}$. The *key implication here is that Lemma 13 provides us a method to find a point in the common set*. An example of Lemma 13 in $\mathbf{E}^2$ is illustrated in Figure 1.

Now, the main question is under what conditions $\bigcap_{p\in[n]} E_p$ is nonempty. Next we show that the nonemptiness of $\bigcap_{p\in[n]} E_p$ can be guaranteed if the diameter of the confidence sets is sufficiently small. This establishes a condition for the number of samples required by the algorithm.

**Theorem 14.** *Given a collection of confident set $\{\mathcal{C}_p\}_{p\in[n]}$ and let $Q = \{x^p\}_{p\in[n]}$, for any $x^p \in \mathcal{C}_p$. For any $p \in [n]$, denote $H_p(Q)$ as the $(n-1)$-dimensional hyperplane with constant $(v^p, c^p)$, $\|v^p\| = 1$ such that*

$$\begin{cases} \langle v^p, x \rangle = c^p + \max_{q\in[n]\setminus p} \text{diam}(\mathcal{C}_p), & \forall x \in Q \setminus x^p. \\ \langle v^p, x^p \rangle < c^p + \max_{q\in[n]\setminus p} \text{diam}(\mathcal{C}_p). \end{cases} \tag{13}$$

*For all $p \in [n]$, if the following holds*

$$\mathcal{D}(\mathcal{C}_p, H_p(Q)) > 2n \left(\max_{q\in[n]\setminus p} \text{diam}(\mathcal{C}_q)\right); \tag{14}$$

*then there exists a common point. In particular, the point $x^\star = \frac{1}{n}\sum_{p\in[n]} x^p$ is a common point.*

Intuitively, Theorem 14 states that if the distance from a confidence set $\mathcal{C}_p$ to the hyperplane $H_p(Q)$ is relatively large compared to the sum of the diameters of all other confidence sets, then the average of any collection of points in the confidence set must be a common point. As such, Theorem 14 determines the stopping condition for Algorithm 1 and provide us a explicit way to find a common point, which validates the correctness of Algorithm 1. In particular, Algorithm 2 checks if conditions (14) are satisfied for the confidence sets in each round. If the conditions are satisfied, then Algorithm 1 stops sampling and returns $x^\star$ as the common point.

Note that while the diameters of confidence sets can be controlled by the number of samples regarding the marginal vector, $\mathcal{D}(\mathcal{C}_p, H_p(Q))$ is a random variable and needs to be handled with care. We show that there exist choices of $n$ vertices such that the simplex formed by them has a sufficiently large width, resulting in the stopping condition being satisfied with high probability after $\text{poly}(n, \varsigma^{-1})$ number of samples.

## 5.2 Sample Complexity Analysis

Now, we show that, the conditions of Theorem 14 can be satisfied with high probability. The distance $\mathcal{D}(\mathcal{C}_p, H_p(Q))$, $p \in [n]$ can be lower bounded by the width of the ground-truth simplex, which is defined as follows:

**Definition 15** (**Width of simplex**). Given $n$ points $\{x^1, ...x^n\}$ in $\mathbb{R}^n$, let matrix $P = [x^i]_{i \in [n]}$, we define the matrix of coordinates of the points in $P$ w.r.t. $x^i$ as $\mathrm{coM}(P, i) := [(x^j - x^i)]_{j \neq i} \in \mathbb{R}^{n \times (n-1)}$. Denote $\sigma_k(M)$ as the $k^{\mathrm{th}}$ singular value of matrix $M$ (with descending order). We define the *width* of the simplex whose coordinate matrix is $P$ as follows

$$\vartheta(P) := \min_{i \in [n]} \sigma_{n-1} \left( \mathrm{coM}(P, i) \right). \tag{15}$$

**Lemma 16.** *Given $n$ points $\{x^1, ..., x^n\}$ in $\mathbb{R}^n$, let $M$ be the matrix corresponding to these points, assume that $0 < M_{ij} < 1$ and $\vartheta(M) \geq \sigma$, for some constant $\sigma > 0$. Let $R \in \mathbb{R}^{n \times n}$ be a perturbation matrix, such that its entries $|R_{ij}| < \epsilon/2$, $\forall(i, j)$, and $0 < \epsilon < \sigma^2/3n^3$. Let $h_{\min}$ be a smallest magnitude of the altitude of the simplex corresponding to the matrix $M + R$. One has that*

$$h_{\min} \geq \sqrt{\sigma^2 - 6n^3 \epsilon}. \tag{16}$$

Lemma 16 guarantees that if the width of the ground truth simplex is relatively large compared to the diameter of the confidence set, then the heights of the estimated simplex are also large. We now provide an example of a collection of permutation orders corresponding to a set of vertices as follows.

**Proposition 17.** *Fix any $\omega \in \mathfrak{S}_n$, consider the collection of permutation $\mathcal{P} = \{\omega, \omega s_1, \ldots, \omega s_{n-1}\}$ and matrix $M = [\phi^{\omega'}]_{\omega' \in \mathcal{P}}$. The width of the simplex that corresponds to $M$, is upper bounded as $\vartheta(M) \geq 0.5\varsigma n^{-3/2}$.*

The vertex set in Proposition 17 comprises one vertex and its $(n - 1)$ adjacent vertices. Combining Lemma 16, Proposition 17 with the stopping condition provided by Theorem 14, we now can guarantee the sample complexity of our algorithm:

**Theorem 18.** *With the choice of collection of permutation order $\mathcal{P}$ as in Proposition 17, and suppose that Assumption 4 holds. Then, for any $\delta \in [0, 1]$, if the number of samples is bounded by*

$$T = O \left( \frac{n^{15} \log(n\delta^{-1} \varsigma^{-1})}{\varsigma^4} \right), \tag{17}$$

*the* `Common-Points-Picking` *algorithm returns a point in E-Core with probability at least $1 - \delta$.*

While the choice of vertices in Proposition 17 achieves polynomial sample complexity, the width of the simplex decreases with dimension growth, hindering its sub-optimality. An alternative choice of vertices is those corresponding to cyclic permutation, denoted as $\mathfrak{C}_n \subset \mathfrak{S}_n$, which have a larger width in large subsets of strictly convex games (as observed in simulations) but can be difficult to verify in the worst case. We refer readers to Appendix A.4 for the detail simulation and discussion on the choice of set of $n$ vertices. Based on this observation, we achieve the sample complexity which better dependence on $n$ as follows.

**Theorem 19.** *Suppose Assumption 4 holds. Let $\mathcal{P} = \mathfrak{S}_n$ the collection of cyclic permutations, and denote the coordinate matrix of the corresponding vertices as $W$. Assume that the width of the simplex $\vartheta(W) \geq \frac{n\varsigma}{c_W}$ for some $c_W > 0$. Then, for any $\delta \in [0, 1]$, if number of samples is*

$$T = O \left( \frac{n^5 c_W^4 \log(nc_W \delta^{-1} \varsigma^{-1})}{\varsigma^4} \right), \tag{18}$$

*the* `Common-Points-Picking` *algorithm returns a point in E-Core with probability at least $1 - \delta$.*

It is worth noting that our algorithm does not require information about the constants of the game $\varsigma$, $c_W$; instead, the number of samples required automatically scales with these constants. This indicates that our algorithm is highly adaptive and requires fewer samples for benign game instances.

**Remark 20** (**Comment on sample complexity lower bounds**). Deriving a lower bound is indeed important, but comes with several significant challenges. E.g., one possible direction is to extend the game instances in Theorem 7. However, there are two key technical issues with this idea: (1) Modifying the face game instance to ensure strict convexity is challenging; (2) It remains unclear how to generalize two face-game instances into $\text{poly}(n)$ game instances such that their cores do not intersect and the statistical distance of the reward can be upper bounded, which is crucial for showing $\text{poly}(n)$ dependencies in the lower bound (we refer the reader to appendix E.2 for further discussions). Given the unresolved challenges, deriving a lower bound remains an open question.

# 6 Experiment

To illustrate the sample complexity of our algorithm in practice and compare it with our theoretical upper bound, we have conducted a simulation as described below. Code is available at: https://github.com/NamTranKekL/ConstantStrictlyConvexGame.git.

**Simulation setting:** We generate convex game of $n$ players with the expected reward function $f$ defined recursively as follows: For each $S \subset N$ s.t. $i \notin S$,

$$f(S \cup \{i\}) = f(S) + |S| + 1 + 0.9\omega.$$

for some $\omega$ sampled i.i.d. from the uniform distribution $\text{Unif}([0, 1])$. We then normalize the value of the reward function within the range $[0, 1]$. It is straightforward to verify that the strict convexity constant is $\varsigma \approx 0.1/n$. From the simulation results in Figure 2 (LHS), we can see that the growth pattern nearly matches that of the theoretical bound given in Theorem 19, indicating that our theoretical bound is highly informative.

Moreover, to demonstrate that our algorithm is robust even when the strict convexity assumption is violated, we ran a simulation where the characteristic function is only convex, i.e., the strict convexity constant is arbitrarily small, as follows:

$$f(S \cup \{i\}) = f(S) + |S| + 1 + \omega.$$

We use the cyclic permutations $\mathfrak{C}_n$ as the input for the algorithm. In Figure 2 (RHS), one can see that the number of samples required as $n$ grows is sub-exponential, indicating that our algorithm is robust when the strict convexity assumption is violated.

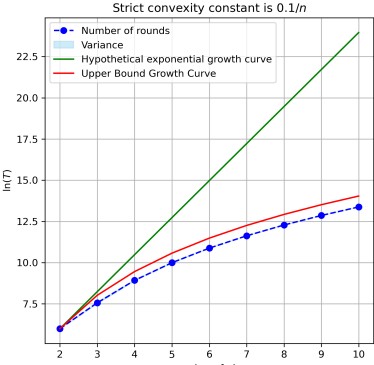 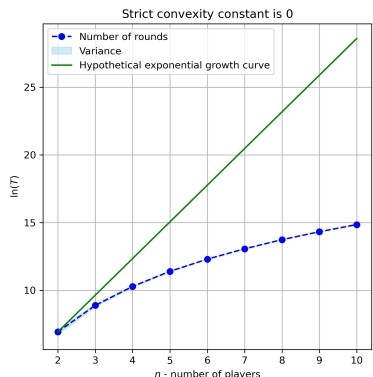

Figure 2: Simulation with game of $n \in \{2, ..., 10\}$ players, where the strict convexity constant $\varsigma$ is $0.1/n$ in the LHS and $0$ in the RHS.

# 7 Conclusion and Future Work

In this paper, we address the challenge of learning the expected core of a strictly convex stochastic cooperative game. Under the assumptions of strict convexity and a large interior of the core, we introduce an algorithm named `Common-Points-Picking` to learn the expected core. Our algorithm guarantees termination after $\text{poly}\left(n, \log(\delta^{-1}), \varsigma^{-1}\right)$ samples and returns a point in the expected core with probability $(1 - \delta)$. For future work, we will investigate whether the sample complexity of our algorithm can be further improved by incorporating adaptive sampling techniques into the algorithm, along with developing a lower bound for the class of games.

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

# Contents of Appendix

## A   Preliminary and Convex Game

### A.1   Proof of Theorem 7

Here and onwards, we adopt the following notation convention: for real numbers $a, b \in [0, 1]$, KL $(a, b)$ represents the KL-divergence KL $(p, q)$ where $p, q$ are probability distributions on $\{0, 1\}$ such that $p(1) = a, \ q(1) = b$. In other words,

$$\text{KL}\,(a, b) = a \ln \left( \tfrac{a}{b} \right) + (1 - a) \ln \left( \tfrac{1-a}{1-b} \right).$$

**Lemma 21** ([16]).  *For any* $0 < \varepsilon < y \leq 1$, KL $(y - \varepsilon, y) < \varepsilon^2 / y(1 - y)$.

Before stating the proof of Theorem 7, let us introduce some extra notations. Given a game $G = (N, \mathbb{P})$, with the expected reward function $\mu$, we define the following.

- $H_C(G) := \{x \in \mathbb{R}^n \mid x(C) = \mu(C)\}$ is the hyperplane corresponding to the effective allocation w.r.t coalition $C$.

- E-Core$(G)$ is the expected core of the game $G$.

- $F_C(G) := \text{E-Core}(G) \cap H_{N \setminus C}(G)$ is facet of the E-Core corresponding to the coalition $C$.

We use the following definition of the face games in Theorem 7, introduced by [14].

**Definition 22** (Face Game).  Given a game $G = (N, \mathbb{P})$ with $\mu(S) = \mathbb{E}_{r \sim \mathbb{P}_S}[r]$, $\forall S \subset N$. For any $C \subset N$, define a face game $G(C) = (N, \mathbb{P}^C)$ with $\mu_{F_C}(S) = \mathbb{E}_{r \sim \mathbb{P}_S^C}[r]$ such that, for any $S \subset N$,

$$\mu_{F_C}(S) = \mu((S \cap C) \cup (N \setminus C)) - \mu(N \setminus C) + \mu(S \cap (N \setminus C)). \tag{19}$$

[14] showed that the expected core of $G(C)$ is exactly the facet of E-Core$(G)$ corresponding $C$, that is, E-Core$(G(C)) = F_C(G)$. As noted in [32], one can decompose the reward function of the face game as follows. For any $S \subset N$, we have that

$$\mu_{F_C}(S) = \mu_{F_C}(S \cap C) + \mu_{F_C}(S \cap (N \setminus C)). \tag{20}$$

We now proceed the proof of Theorem 7.

***Proof of Theorem*** 7 . Denote the set convex games with Bernolli reward as **GB**, that is,

$$\mathbf{GB} = \{G = (N, \mathbb{P}) \mid \mathbb{P} = \{\mathbb{P}_S\}_{S \subseteq N}; \ \mathbb{P}_S \in \mathcal{M}(\{0,1\}), \ \forall S \subseteq N\}.$$

**Face-game instances and the distance between their E-Core.** We first define two games, $G_0$ and $G_1$, with a full-dimensional E-Core, such that $G_1$ is a slight perturbation of $G_0$. Next, we define face games corresponding to $G_0$ and $G_1$ using the perturbed facet. We then show that the distance between the cores of these two face games is at least some positive number $\varepsilon > 0$.

Define a strictly convex game $G_0 := (N, \mathbb{P}) \in \mathbf{GB}$, such that $\mu^0(S) := \mathbb{E}_{r \sim \mathbb{P}_S}[r]$, and assume that $\mu^0$ is $\varsigma$-strictly supermodular. Now, fix a subset $C \subset N$, let define a perturbed game instance $G_1 := (N, \mathbb{Q}) \in \mathbf{GB}$, with $\mu^1(S) := \mathbb{E}_{r \sim \mathbb{Q}_S}[r]$ such that

$$\begin{cases} \mu^1(C) := \mu^0(C) - \varepsilon; \\ \mu^1(S) := \mu^0(S); & \forall S \subset N, \ S \neq C; \end{cases} \tag{21}$$

for some $0 < \varepsilon < \varsigma$. It is straightforward that $G_1$ is $(\varsigma - \varepsilon)$-strictly convex. Therefore, E-Core($G_0$) and E-Core($G_1$) are both full-dimensional.

Fixing a coalition $C \subset N$, we now construct the face games from $G_0$, $G_1$ as in Definition 22. Let $G_0(C) := (N, \mathbb{P}^C)$, $G_1(C) := (N, \mathbb{Q}^C) \in \mathbf{GB}$, whose expected rewards $\mu^0_{F_C}$ and $\mu^1_{F_C}$ are defined by applying (19) to $\mu^0$ and $\mu^1$ respectively. Now, we consider the difference between the expected reward function of these two games.

$$\begin{cases} |\mu^1_{F_C}(S) - \mu^0_{F_C}(S)| = 0 & \forall S \subset N \setminus C \\ |\mu^1_{F_C}(S) - \mu^0_{F_C}(S)| = \varepsilon & \forall S \subseteq C \\ |\mu^1_{F_C}(N \setminus C) - \mu^0_{F_C}(N \setminus C)| = \varepsilon. \end{cases} \tag{22}$$

As one can always decompose the set $S = (S \cap C) \cup (S \cap N \setminus C)$, by the decomposibility of the face game (20), we has that

$$|\mu^1_{F_C}(S) - \mu^0_{F_C}(S)| \leq \varepsilon, \ \forall S \subset N. \tag{23}$$

As the core of face game $G_0(C)$ and $G_1(C)$ lie on the hyperplane corresponding to the coalition $N \setminus C$, and the distance between the hyperplanes of $G_0$ and $G_1$ is $\varepsilon$, which lower bounds the distance between the expected core of $G_0(C)$ and $G_1(C)$. In particular, as E-Core($G_0(C)$) = $F_C(G_0)$ and E-Core($G_1(C)$) = $F_C(G_1)$, and $|\mu^1(N \setminus C) - \mu^0(N \setminus C)| = \varepsilon$, which leads to $\mathcal{D}(H_{N \setminus C}(G_0), H_{N \setminus C}(G_1)) = \varepsilon$, we have that

$$\mathcal{D}\left(\text{E-Core}(G_0(C)), \text{E-Core}(G_1(C))\right) \geq \varepsilon. \tag{24}$$

**The KL distance and imposibility of learning low-dimensional E-Core.** We show that, with probability $\delta \in (0, 0.2)$, any learner cannot distinguish between $G_0(C)$ and $G_1(C)$ given there are finite number of samples. We use the information-theoretic framework similar which is well developed within multi-armed bandit literature.

We first upper bound the KL-distance between $\mathbb{P}^C_S, \mathbb{Q}^C_S$, $\forall S \subset N$. Denote $c_1 := \min_{S \subset N} \left(\mu^0_{F_C}(S)(1 - \mu^0_{F_C}(S))\right) > 0$, by Lemma 21, we have that

$$\text{KL}\left(\mathbb{P}^C_S, \mathbb{Q}^C_S\right) = \text{KL}\left(\mu^0_{F_C}(S), \mu^1_{F_C}(S)\right) \leq \frac{\varepsilon^2}{c_1}, \quad \forall S \subset N.$$

Define the probability space $\Psi = 2^N \times \{0, 1\}$. Fix any algorithm (possibly randomised) $\mathcal{A}$. At round $t$, denote $(S_t, r_t) \in \Psi$ as the coalition selected by the algorithm and the reward return by the environment. At round $s < t$, denote $\nu^t_0$, $\nu^t_1$ as the probability distribution over $\Psi^t$ determined by $\mathcal{A}$ and $\mathbb{P}$, $\mathbb{Q}$ accordingly.

We have the following, as stated in the appendix of [16]. For any $u < t$, one has that,

$$
\mathrm{KL}\left(\nu_0^u, \nu_1^u\right) = \sum_{\psi^{u-1} \in \Psi^{u-1}} \nu_0^u(\psi^u) \log \left(\frac{\nu_0^u(\psi^u \mid \psi^{u-1})}{\nu_1^u(\psi^u \mid \psi^{u-1})}\right)
$$

$$
= \sum_{\psi^{u-1} \in \Psi^{u-1}} \nu_0^u(\psi^u) \log \left(\frac{\nu_0^u(S_u \mid \psi^{u-1})}{\nu_1^u(S_u \mid \psi^{u-1})} \cdot \frac{\nu_0^u(r_u \mid S_u, \psi^{u-1})}{\nu_1^u(r_u \mid S_u, \psi^{u-1})}\right)
$$

$$
= \sum_{\psi^{u-1} \in \Psi^{u-1}} \nu_0^u(\psi^u) \log \left(\frac{\nu_0^u(r_u \mid S_u, \psi^{u-1})}{\nu_1^u(r_u \mid S_u, \psi^{u-1})}\right)
$$

[As the distribution of $S_u$ depends only on $\mathcal{A}$, not on the distribution $\nu_0^t$, $\nu_1^t$.]

$$
= \sum_{\psi^{u-1} \in \Psi^{u-1}} \sum_{S_u \in 2^N} \sum_{r_u \in \{0,1\}} \nu_0^u(r_u \mid S_u, \psi^{u-1}) \log \left(\frac{\nu_0^u(r_u \mid S_u, \psi^{u-1})}{\nu_1^u(r_u \mid S_u, \psi^{u-1})}\right) \nu_0^u(S_u, \psi^{u-1})
$$

$$
= \sum_{\psi^{u-1} \in \Psi^{u-1}} \sum_{S_u \in 2^N} \mathrm{KL}\left(\mu_{F_C}^0(S_u), \mu_{F_C}^1(S_u)\right) \nu_0^u(S_u, \psi^{u-1})
$$

$$
\leq \frac{\varepsilon^2}{c_1}.
$$

The last inequality hold because $\mathrm{KL}\left(\mu_{F_C}^0(S), \mu_{F_C}^1(S)\right) \leq \frac{\varepsilon^2}{c_1}$, $\forall S \in 2^N$.

We have that

$$
\mathrm{KL}\left(\nu_0^t, \nu_1^t\right) = \sum_{u=1}^t \mathrm{KL}\left(\nu_0^u, \nu_1^u\right) \leq \frac{t\varepsilon^2}{c_1}. \tag{25}
$$

As we can choose $\varepsilon$ to be arbitrarily small, we can choose $\varepsilon$ such that $\mathrm{KL}\left(\nu_0^t, \nu_1^t\right) \leq 0.1$.

Now, define the event $\mathcal{E}$ as the event that $\mathcal{A}$ outputs a point in E-Core$(G_0(C))$, assume that $\nu_0^t(\mathcal{E})$ with probability at least $0.8$. Note that, as E-Core$(G_0(C)) \cap$ E-Core$(G_1(C)) = \varnothing$, $\mathcal{E}$ represents the event where the algorithm fails to output a stable allocation with the game instance $G_1(C)$. We have that from [16]'s Lemma A.5,

$$
\nu_1^t(\mathcal{E}) \geq \nu_0^t(\mathcal{E}) \exp \left(-\frac{\mathrm{KL}\left(\nu_0^t, \nu_1^t\right) + 1/e}{\nu_0^t(\mathcal{E})}\right) > 0.8 \exp \left(-\frac{0.1 + 1/e}{0.8}\right) > 0.3. \tag{26}
$$

As it holds for any $t > 0$, this means that for any finite number of samples, with probability at least $0.1$, the algorithm will output the incorrect point. $\qquad\square$

**Remark 23.** Upon closely examining the face game instances in the proof of Theorem 7, we can see that even if the E-Core is full-dimensional, but the width of the interior of E-Core is arbitrarily small, it is still not possible to learn a point in E-Core with high probability and finite samples. To see this, let us create two perturbed game instances of the face game instances such that their E-Core are full-dimensional but have an arbitrarily narrow interior. The construction can be done by applying modification of equation (19) with an arbitrarily small constant $\zeta > 0$ on the game $\tilde{G}_0$, $\tilde{G}_1$, as follows.

$$
\mu_{F_C}(S) = \mu((S \cap C) \cup (N \setminus C)) - \mu(N \setminus C) + \mu(S \cap (N \setminus C)) + \zeta. \tag{27}
$$

Now, as long as $\zeta < \varepsilon/2$, meaning that the width of the interior of their E-Core is less than half of the distance between the original face games, the distance between their E-Core remains positive. Therefore, as the KL distance between the reward distributions is arbitrarily small but the two games do not share any common stable allocation, no algorithm can output a stable allocation of the ground truth game with high probability and finite samples.

## A.2 E-Core of convex games and Generalised Permutahedra

Formulating the coordinates of the vertices of the core can be achieved using the connection between the core of a convex game and the generalised permutahedron. There is an equivalence between

generalised permutahedra and polymatroids; it was also shown in [25] that the core of each convex game is a generalised permutahedron.

For any $\omega \in \mathfrak{S}_n$, let $\mathbf{I}^\omega = (\omega(1), ..., \omega(n))$. The $n$-permutahedron is defined as $\text{Conv}(\{\mathbf{I}^\omega \mid \omega \in \mathfrak{S}_n\})$. A generalised permutahedron can be defined as a deformation of the permutahedron, that is, a polytope obtained by moving the vertices of the usual permutohedron so that the directions of all edges are preserved [21]. Formally, the edge of the core corresponding to adjacent vertices $\phi^\omega$, $\phi^{\omega s_i}$ can be written as

$$\phi^\omega - \phi^{\omega s_i} = k_{\omega,i}(e_{\omega(i)} - e_{\omega(i+1)}), \tag{28}$$

Where, $k_{\omega,i} \geq 0$, and $e_1, \ldots e_n$ are the coordinate vectors in $\mathbb{R}^n$. If the game is $\varsigma$-strictly convex, $k_{\omega,i} > \varsigma$.

### A.3  Proof of Proposition 17

We utilise the formulation of edges of the generalized permutahedron as described in Subsection A.2 to calculate the matrix of coordinates for the vertices of E-Core. Based on the matrix of coordinates, we now state the proof of Proposition 17.

***Proof of Proposition 17.*** As the set of vertices is $\phi^\omega$ and its $n-1$ neighbors, there are only two cases to consider. First, we need to consider the matrix created by using $\phi_\omega$ as the reference, that is $\text{coM}(M, 1)$. As the neighbors have the same roles, bounding the width of the matrices using any neighbor as a reference point can be done identically. Therefore, we will prove the theorem for $\text{coM}(M, 2)$, and the proof for $\text{coM}(M, i)$, $i \neq 1$ can be done in the same manner. Let us denote

$$V = \text{coM}(M, 1) = \begin{bmatrix} c_1 & 0 & 0 & \cdots & 0 & 0 \\ -c_1 & c_2 & 0 & \cdots & 0 & 0 \\ 0 & -c_2 & c_3 & \cdots & 0 & 0 \\ \vdots & \vdots & \vdots & \ddots & \vdots & \vdots \\ \vdots & \vdots & \vdots & \ddots & \vdots & \vdots \\ 0 & 0 & 0 & \cdots & -c_{n-2} & c_{n-1} \\ 0 & 0 & 0 & \cdots & 0 & -c_{n-1} \end{bmatrix} \in \mathbb{R}^{n \times (n-1)}, \tag{29}$$

$$U = \text{coM}(M, 2) = \begin{bmatrix} -c_1 & -c_1 & -c_1 & -c_1 & \cdots & -c_1 & -c_1 \\ c_1 & c_1 + c_2 & c_1 & c_1 & \cdots & c_1 & c_1 \\ 0 & -c_2 & c_3 & 0 & \cdots & 0 & 0 \\ 0 & 0 & -c_3 & c_4 & \cdots & 0 & 0 \\ \vdots & \vdots & \vdots & \vdots & \ddots & \vdots & \vdots \\ \vdots & \vdots & \vdots & \vdots & \ddots & \vdots & \vdots \\ 0 & 0 & 0 & 0 & \cdots & -c_{n-2} & c_{n-1} \\ 0 & 0 & 0 & 0 & \cdots & 0 & -c_{n-1} \end{bmatrix} \in \mathbb{R}^{n \times (n-1)}, \tag{30}$$

in which each $c_i > \varsigma$.

We will exploit the following norm inequality in the proof. For any $A_1, \ldots, A_n \in \mathbb{R}$, we use the following inequality (norm 2 vs. norm 1 of vectors)

$$\sum_{i=1}^n A_i^2 \geq \frac{\left(\sum_{i=1}^n A_i\right)^2}{n} \tag{31}$$

**Consider V.** Consider a unit vector $x = (x_1, ..., x_{n-1})$. We have

$$Vx = \begin{bmatrix} c_1 x_1 \\ -c_1 x_1 + c_2 x_2 \\ -c_2 x_2 + c_3 x_3 \\ \cdots \\ -c_{n-2} x_{n-2} + c_{n-1} x_{n-1} \\ -c_{n-1} x_{n-1} \end{bmatrix} \tag{32}$$

Applying the Ineq. (31) for $A_1 = c_1 x_1$, $A_2 = -c_1 x_1 + c_2 x_2$, $A_{n-1} = -c_{n-2} x_{n-2} + c_{n-1} x_{n-1}$, $A_n = -c_{n-1} x_{n-1}$ gives

$$\|Vx\|^2 \geq \frac{c_1^2 x_1^2}{n} \geq \frac{\varsigma^2 x_1^2}{n};$$

$$\|Vx\|^2 \geq c_1^2 x_1^2 + (-c_1 x_1 + c_2 x_2)^2 \geq \frac{c_2^2 x_2^2}{n} \geq \frac{\varsigma^2 x_2^2}{n};$$

$$\cdots$$

$$\|Vx\|^2 \geq \frac{\varsigma^2 x_{n-1}^2}{n}. \tag{33}$$

Therefore,

$$n\|Vx\|^2 \geq \frac{\varsigma^2 (x_1^2 + \cdots + x_{n-1}^2)}{n} = \frac{\varsigma^2}{n} \tag{34}$$

Therefore $\|Vx\| \geq \varsigma/n$, hence $\sigma_{n-1}(V) \geq \varsigma/n$.

**Consider U.** Similarly, consider a unit vector $x = (x_1, ..., x_{n-1})$. We have

$$Ux = \begin{bmatrix} -c_1(x_1 + x_2 + ... + x_{n-1}) \\ c_1(x_1 + x_2 + ... + x_{n-1}) + c_2 x_2 \\ -c_2 x_2 + c_3 x_3 \\ -c_3 x_3 + c_4 x_4 \\ \cdots \\ -c_{n-2} x_{n-2} + c_{n-1} x_{n-1} \\ -c_{n-1} x_{n-1} \end{bmatrix} \tag{35}$$

Applying the Ineq. (31) for $A_1 = c_1(x_1 + x_2 + ... + x_{n-1})$, $A_2 = c_1(x_1 + x_2 + ... + x_{n-1}) + c_2 x_2$, $A_3 = -c_2 x_2 + c_3 x_3$, $A_4 = -c_3 x_3 + c_4 x_4$, ..., $A_{n-1} = -c_{n-2} x_{n-2} + c_{n-1} x_{n-1}$, $A_n = -c_{n-1} x_{n-1}$ gives

Note that

$$\|Ux\|^2 \geq \frac{\varsigma^2 (x_1 + x_2 + ... + x_{n-1})^2}{n};$$

$$\|Ux\|^2 \geq c_1^2(x_1 + x_2 + ... + x_{n-1})^2 + (c_1(x_1 + x_2 + ... + x_{n-1}) + c_2 x_2)^2 \geq \frac{c_2^2 x_2^2}{n} \geq \frac{\varsigma^2 x_2^2}{n};$$

$$\|Ux\|^2 \geq c_1^2(x_1 + x_2 + ... + x_{n-1})^2 + (c_1(x_1 + x_2 + ... + x_{n-1}) + c_2 x_2)^2 + (-c_2 x_2 + c_3 x_3)^2 \geq \frac{\varsigma^2 x_3^2}{n};$$

$$\cdots$$

$$\|Ux\|^2 \geq \frac{\varsigma^2 x_{n-1}^2}{n} \tag{36}$$

Therefore, we also have

$$n\|Ux\|^2 \geq \frac{\varsigma^2((x_1 + x_2 + ... + x_{n-1})^2 + x_2^2 + ... + x_{n-1}^2)}{n} \geq \frac{\varsigma^2 x_1^2}{n^2} \tag{37}$$

From that, we have that

$$2n\|Ux\|^2 \geq \varsigma^2 \frac{x_1^2}{n^2} + \frac{x_2^2}{n} + \cdots + \frac{x_{n-1}^2}{n} \geq \frac{x_1^2 + ... + x_{n-1}^2}{n^2} = \frac{\varsigma^2}{n^2}, \text{ as } \|x\| = 1 \tag{38}$$

That is, $\|Ux\| \geq \frac{\varsigma^2}{\sqrt{2n^3}}$. Therefore, $\sigma_{n-1}(U) \geq \frac{\varsigma^2}{\sqrt{2n^3}}$.

Therefore, we have that $\vartheta(M) > \frac{\varsigma^2}{\sqrt{2n^3}}$. $\qquad\square$

## A.4 Alternative choice of $n$ vertices of E-Core

In this subsection, we provide an alternative choice of vertices rather than that in Proposition 17. Recall that, with the choice of vertices in Proposition 17, the lower bound for the width of the simplex

diminishes when the dimension increases. This leads to a large dependence of the sample complexity on $n$. To mitigate this, we investigate other choices of $n$ vertices. To see this, we first recall the equivalence between E-Core and generalized permutahedra as explained in Subsection A.2.

However, even in the case of a simple permutahedron, if the set of vertices is not carefully chosen, the width of their convex can be proportionally small w.r.t. $n$, as demonstrated in the next proposition. In particular, the same choice of vertices as in 17 results in the simplex with diminishing width as follows.

**Proposition 24.** *Consider a permutahedron, fix $\omega \in \mathfrak{S}_n$, consider the matrix $W = [\phi^\omega, \mathbf{I}^{\omega s_1}, \mathbf{I}^{\omega s_2}, \dots, \mathbf{I}^{\omega s_{n-1}}]$. The width of the simplex that corresponds to $M$, is upper bounded as follows:*

$$\vartheta(M) \leq \frac{3}{n}. \tag{39}$$

*Proof.* The coordinate matrix w.r.t. $\phi^\omega$, that is, $\text{coM}(M, 1)$ can be written as follows.

$$V = \begin{bmatrix} 1 & 0 & 0 & \cdots & 0 & 0 \\ -1 & 1 & 0 & \cdots & 0 & 0 \\ 0 & -1 & 1 & \cdots & 0 & 0 \\ \vdots & \vdots & \vdots & \ddots & \vdots & \vdots \\ \vdots & \vdots & \vdots & \ddots & \vdots & \vdots \\ 0 & 0 & 0 & \cdots & -1 & 1 \\ 0 & 0 & 0 & \cdots & 0 & -1 \end{bmatrix} \in \mathbb{R}^{n \times (n-1)} \tag{40}$$

Therefore, the Gram matrix is

$$G := V^\top V = \begin{bmatrix} 2 & -1 & 0 & 0 & 0 & \cdots & 0 & 0 & 0 \\ -1 & 2 & -1 & 0 & 0 & \cdots & 0 & 0 & 0 \\ 0 & -1 & 2 & -1 & 0 & \cdots & 0 & 0 & 0 \\ \vdots & \vdots & \vdots & \vdots & \vdots & \ddots & \vdots & \vdots & \vdots \\ \vdots & \vdots & \vdots & \vdots & \vdots & \ddots & \vdots & \vdots & \vdots \\ \vdots & \vdots & \vdots & \vdots & \vdots & \ddots & \vdots & \vdots & \vdots \\ \vdots & \vdots & \vdots & \vdots & \vdots & \ddots & \vdots & \vdots & \vdots \\ 0 & 0 & 0 & \dots & 0 & 0 & -1 & 2 & -1 \\ 0 & 0 & 0 & \dots & 0 & 0 & 0 & -1 & 2 \end{bmatrix} \in \mathbb{R}^{(n-1) \times (n-1)}. \tag{41}$$

Note that $G$ is a tridiagonal matrix and also Toeplitz matrix, therefore, its minimum eigenvalues has closed form as follows

$$\lambda_{n-1}(G) = 2 + 2\cos\left(\frac{(n-1)\pi}{n}\right) = 2\sin^2\left(\frac{\pi}{2n}\right) \leq \frac{5}{n^2}; \tag{42}$$

as $\left|\sin\left(\frac{\pi}{2n}\right)\right| \leq \frac{\pi}{2n}$. Therefore, $\vartheta(M) \leq \sigma_{n-1}(V) = \sqrt{\lambda_{n-1}(G)} \leq \frac{3}{n}$. $\qquad\square$

Proposition 24 highlights the challenge of selecting a set of vertices such that the width does not contract with the increasing dimension, even in the case of a simple permutahedron. Denote $\mathfrak{C}_n \subset \mathfrak{S}_n$ as the group of cyclic permutations of length $n$. One potential candidate for such a set of vertices is the collection corresponding to cyclic permutations $\mathfrak{C}_n$, as described in the next proposition.

**Proposition 25.** *Consider the matrix $\overline{W} = [\mathbf{I}^\omega]_{\omega \in \mathfrak{C}_n}$. We have that*

$$\vartheta(\overline{W}) \geq \frac{n}{2}. \tag{43}$$

*Proof.* The form of matrix $\overline{W}$ is as follows

$$\overline{W} = \begin{bmatrix} 1 & n & n-1 & \dots & 2 \\ 2 & 1 & n & \dots & 3 \\ 3 & 2 & 1 & \dots & 4 \\ \vdots & \vdots & \vdots & \vdots & \vdots \\ n-1 & n-2 & n-3 & \dots & n-1 \\ n & n-1 & n-2 & \dots & 1 \end{bmatrix}. \tag{44}$$

The coordinate matrix w.r.t. the first column is as follows

$$V = \mathrm{coM}(\overline{W}, 1) = \begin{bmatrix} n-1 & n-2 & \dots & 1 \\ -1 & n-2 & \dots & 1 \\ -1 & -2 & \dots & 1 \\ \vdots & \vdots & \ddots & \vdots \\ -1 & -2 & \dots & 1 \\ -1 & -2 & \dots & -(n-1) \end{bmatrix}. \tag{45}$$

Let $u \in \mathbb{R}^{n-1}$ be any unit vector, and let $z = Vu \in \mathbb{R}^n$. We have that

$$z_i - z_{i+1} = nu_i. \tag{46}$$

Let us consider

$$\begin{aligned} 4\|z\|^2 &= 4z_1^2 + 4z_2^2 + \cdots + 4z_n^2 \\ &= 2z_1^2 + [(z_1 + z_2)^2 + (z_1 - z_2)^2] + [(z_2 + z_3)^2 + (z_2 - z_3)^2] \\ &\quad + \cdots + [(z_{n-1} + z_n)^2 + (z_{n-1} - z_n)^2] + 2z_n^2 \\ &\geq (z_1 - z_2)^2 + (z_2 - z_3)^2 + \cdots + (z_{n-1} - z_n)^2 \\ &= n^2(u_1^2 + u_2^2 + \cdots + u_{n-1}^2) = n^2. \end{aligned} \tag{47}$$

Therefore, we have that

$$\sigma_{n-1}(V) = \min_{u:\|u\|=1} \sqrt{\frac{\|Vu\|^2}{\|u\|^2}} \geq \frac{n}{2}. \tag{48}$$

It is straightforward that if one takes any column of $\overline{W}$ as a reference column, the resulting coordinate matrices have identical singular values. In particular, for any $i, j \in [n]$

$$\mathrm{coM}(\overline{W}, i) = P \cdot \mathrm{coM}(\overline{W}, j),$$

where $P$ is a permutation matrix, thus, their singular values are identical. Therefore, we have that

$$\vartheta(\overline{W}) \geq \frac{n}{2}.$$

$\square$

As a result, the set of vertices corresponding to cyclic permutations is a sensible choice. In case of a generalised permutahedron, let us define

$$W := [\phi^\omega]_{\omega \in \mathfrak{C}_n}. \tag{49}$$

As generalised permutahedra are deformations of the permutahedron, we expect that $\vartheta(W)$ is reasonably large for a broad class of strictly convex games. In particular, we consider the class of strictly convex games in which the width $\vartheta(W)$ is lower bounded, as in the following assumption:

**Assumption 26.** *The width of the simplex that corresponds to $W$ in* (49) *is bounded as follows*

$$\vartheta(W) \geq \frac{n\varsigma}{c_W}, \tag{50}$$

*for some constant $c_W > 0$.*

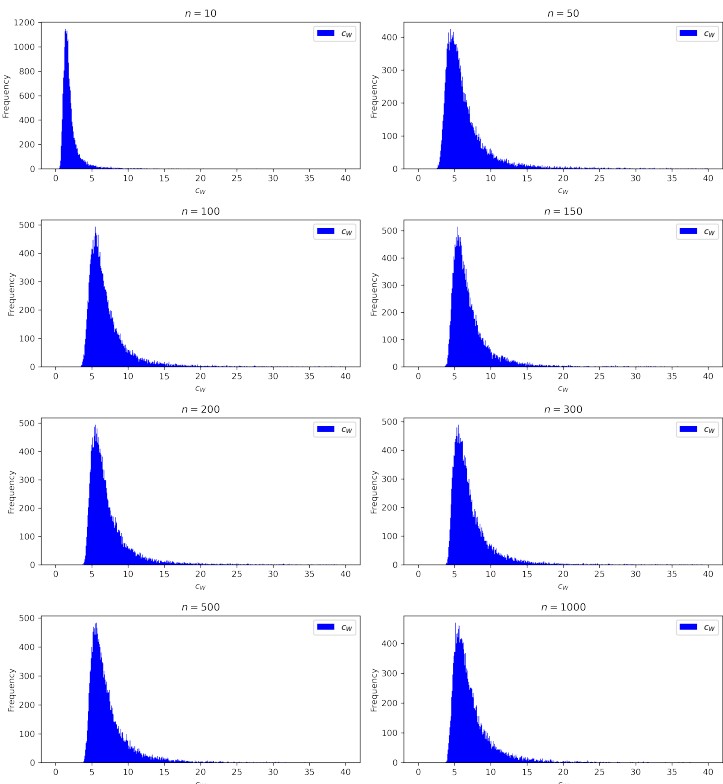

Figure 3: $c_W$ with $n \in \{10,\ 50,\ 100,\ 150,\ 200,\ 300,\ 500,\ 1000\}, \varsigma = \frac{0.1}{n}$, and 20000 trials

These parameters will eventually play a crucial role in determining the number of samples required using this choice of $n$ permutation orders. Although proving an exact upper bound for $c_W$ in all strictly convex games is challenging, we conjecture that $c_W$ is relatively small in a large subset of the games.

To investigate Assumption 26, we conducted a simulation to compute the constant $c_W$ of the minimum singular value $\sigma_{n-1}(M)$. For each case where $n$ takes values of $(10,\ 50,\ 100,\ 150,\ 200,\ 300,\ 500,\ 1000)$, the simulation consisted of 20000 game trials with $\varsigma = 0.1/n$. As depicted in Figure 3, the values of $c_W$ tend to be relatively small and highly concentrated within the interval $(0, 30)$. This observation suggests that for most cases of strictly convex games, $c_W$ remains reasonably small. Consequently, our algorithm exhibits relatively low sample complexity. Code of the experiment is available at: `https://github.com/NamTranKekL/ConstantStrictlyConvexGame.git`.

For each case where $n$ takes values of $(10,\ 50,\ 100,\ 150,\ 200,\ 300,\ 500,\ 1000)$, the simulation consisted of 20000 game trials with $\varsigma = 0.1/n$. As depicted in Figure 3, the values of $c_W$ tend to be relatively small and highly concentrated within the interval $(0, 30)$. This observation suggests that for most cases of strictly convex games, $c_W$ remains reasonably small. The results indicate that $c_W$ tends to be relatively small with high probability, and does not depend on the value of $n$.

## B  On the Stopping Condition

***Proof of Proposition 5***. For each $\mathcal{C}_p$, choose a point in its interior, denote as $x^p$. As there are at most $n-1$ points $\{x^p\}_{p \in [|\mathcal{P}|]}$, there exists a $(n-2)$-dimensional hyperplane $H$ that contains $\{x^p\}_{p \in [|\mathcal{P}|]}$. Let $\tilde{H}$ be a hyperplane parallel to $H$ and let the distance $\mathcal{D}(H, \tilde{H})$ be arbitrary small.

As confidence sets are full-dimensional $(n-1)$, $\tilde{H}$ must also intersect with the interiors of all confidence sets. Since $H$ and $\tilde{H}$ are parallel, any convex hull of points within $H$ and $\tilde{H}$ cannot intersect. Therefore, there is no common point. $\qquad\square$

***Proof of Proposition 9.*** The proof spirit is similar to that of Proposition 5.

Let $H$ be the $(n-2)$-dimensional hyperplane that intersects with the interiors of all confidence sets. Let $\tilde{H}$ be a hyperplane parallel to $H$ and let the distance $\mathcal{D}(H, \tilde{H})$ be arbitrary small.

As confidence sets are full-dimensional, $\tilde{H}$ must also intersect with the interiors of all confidence sets. Since $H$ and $\tilde{H}$ are parallel, any convex hull of points within $H$ and $\tilde{H}$ cannot intersect. Therefore, there is no common point. $\qquad\square$

The proof of Theorem 11 is a combination of the classic hyperplane separation theorem and the following lemma.

**Lemma 27.** *Let $\{\mathcal{C}_p\}_{p\in[n]}$ be mutually disjoint compact and convex subsets in $\mathbf{E}^{n-1}$. Suppose there does not exist a $(n-2)$-dimensional hyperplane that intersects with all confidence sets $\mathcal{C}_p$, $\forall p \in [n]$, then for each $p \in [n]$*

$$\mathcal{C}_p \cap \mathrm{Conv}\left(\bigcup_{q\neq p} \mathcal{C}_q\right) = \varnothing. \tag{51}$$

*Proof.* We prove this lemma by contra-position, that is, if there is $\mathcal{C}_p$ such that

$$\mathcal{C}_p \cap \mathrm{Conv}\left(\bigcup_{p\neq q} \mathcal{C}_q\right) \neq \varnothing;$$

then there exist a hyperplane that intersects with all the $\mathcal{C}_p$, $\forall p \in [n]$.

**First**, assume there is a point $x = \mathcal{C}_p \cap \mathrm{Conv}\left(\bigcup_{q\neq p} \mathcal{C}_p\right)$. By Carathéodory's theorem, there are at most $n$ points $x^k \in \bigcup_{q\neq p} \mathcal{C}_q$ such that

$$x = \sum_{k\in[n]} \alpha_k x^k. \tag{52}$$

As each $x^k \in \mathcal{C}_q$ for some $\mathcal{C}_q$, one can rewrite the equation above as

$$x = \sum_{q\neq p} \sum_{k:\, x^k \in \mathcal{C}_q} \alpha_k x^k. \tag{53}$$

Furthermore, we can write

$$\sum_{x^k \in \mathcal{C}_q} \alpha_k x^k = \tilde{\alpha}_q \tilde{x}^q, \quad \text{in which,} \quad \tilde{x}^q := \frac{\sum_{k:\, x^k \in \mathcal{C}_q} \alpha_k x^k}{\sum_{k:\, x^k \in \mathcal{C}_q} \alpha_k}, \quad \text{and} \quad \tilde{\alpha}_q := \sum_{k:\, x^k \in \mathcal{C}_q} \alpha_k. \tag{54}$$

Since $\mathcal{C}_q$ is convex, $\tilde{x}^q \in \mathcal{C}_q$. Substituting (54) into (52), one obtains

$$x = \sum_{q\neq p} \tilde{\alpha}_q \tilde{x}^q. \tag{55}$$

Define $H$ as a hyperplane that passes through all $\tilde{x}_q$, we have that $x \in H$.

**Second**, we now show how to construct a hyperplane that intersects with all $\mathcal{C}_m$, $m \in [n]$. Let $I$ be the set of indices such that $\mathcal{C}_q \ni \tilde{x}_q$. We have two following cases.

    (i) First, if $|I| = n-1$, then $H$ is the $(n-2)$-dimensional hyperplane that intersect with all $\mathcal{C}_m$, $m \in [n]$.

    (ii) Second, if $|I| < n-1$, for any $\mathcal{C}_{q'} \neq \mathcal{C}_p$ that does not contain any $\tilde{x}^q$, we choose any arbitrary point $x^{q'} \in \mathcal{C}_{q'}$. As there are $n-1$ points of $\tilde{x}^q$ and $x^{q'}$, there exists a hyperplane $\overline{H}$ that contains all these points. Furthermore, $\overline{H}$ must contain $x$, so it is the $(n-2)$-dimensional hyperplane that intersects with all sets $\mathcal{C}_m$, $\forall m \in [n]$.

$\square$

Now, we state the proof of Theorem 11.

**Proof of Theorem 11.** As a result of Lemma 27, we have that for all $\mathcal{C}_p, \forall p \in [n]$,

$$\mathcal{C}_p \cap \mathrm{Conv}\left(\bigcup_{q \neq p} \mathcal{C}_q\right) = \varnothing. \tag{56}$$

Therefore, by the hyperplane separation theorem, there must exist a hyperplane that separates $\mathcal{C}_p$ and $\mathrm{Conv}\left(\bigcup_{q \neq p} \mathcal{C}_q\right)$.

$\square$

**Proof of Lemma 13.** Let us denote $\Delta_n$ as $\mathrm{Conv}\left(\{x^p\}_{p \in [n]}\right)$. As there is no hyperplane of dimension $n - 2$ go through all the set $\mathcal{C}_p$, the simplex $\Delta_n$ is $(n-1)$ dimensional. We have that

$$\bigcap_{p \in [n]} E_p \subseteq \Delta_n \iff \Delta_n^c \subseteq \bigcup_{p \in [n]} E_p^c;$$

where $E_p^c$ is the complement of the set $E_p$.

We will prove the RHS of the above. Consider $\hat{x} \in \Delta_n^c$, as $\Delta_n$ is full dimensional, $\hat{x}$ can be uniquely written as affine combination of the vertices, that is,

$$\hat{x} = \sum_{p \in [n]} \lambda_p x^p, \quad \sum_{p \in [n]} \lambda_p = 1.$$

As $\hat{x} \in \Delta_n^c$, there must exist some $\lambda_k < 0$.

Now, we shall prove $\hat{x} \in E_k^c$. Consider the following,

$$\begin{aligned} \langle v^k, \hat{x} \rangle = \left\langle v^k, \sum_{p \in [n]} \lambda_p x^p \right\rangle &= \lambda_k \langle v^k, x^k \rangle + \sum_{p \neq k} \lambda_p \langle v^k, x^p \rangle \\ &> \lambda_k c^k + c^k \sum_{p \neq k} \lambda_p \\ &= c^k \end{aligned} \tag{57}$$

The above inequality holds since $\langle v^k, x^k \rangle < c_k$ and $\lambda_k < 0$. Therefore, $\hat{x} \in E_k^c$. This means that

$$\Delta_n^c \subseteq \bigcup_{k \in [n]} E_k^c. \tag{58}$$

$\square$

**Proof of Theorem 14.** Before proceeding the main proof, we show two simple consequences of the construction of $H_p(Q)$, $p \in [n]$ and the assumption (14).

Fact 1: *Consider $p \in [n]$, $H_p(Q)$ acts as a separating hyperplane for $\mathcal{C}_p$.* To see this, assume that $H_p(Q)$ is not a separate hyperplane for $\mathcal{C}_p$, then there exists $z^p \in \mathcal{C}_p$ such that $\langle v^p, z^p \rangle \geq c^p$. From (13), we have $\langle v^p, x^p \rangle \leq c^p + \max_{q \in [n] \backslash p} \mathrm{diam}(\mathcal{C}_p)$. Then, there are two cases. First, assume that $\langle v^p, x^p \rangle \leq c^p$. As $x^p$, $z^p \in \mathcal{C}_p$ and $\langle v^p, z^p \rangle \geq c^p$, there must exist a point $x$ in the line segment $[x^p, z^p]$ such that $\langle v^p, x \rangle = c^p$. This means that $\mathcal{D}(\mathcal{C}_p, H_p) = 0$, which violates assumption (14). Second, assume that $c^p \leq \langle v^p, x^p \rangle \leq c^p + \max_{q \in [n] \backslash p} \mathrm{diam}(\mathcal{C}_p)$. Then, we have that

$$\mathcal{D}(\mathcal{C}_p, H_p) \leq \mathcal{D}(x^p, H_p) = |\langle v^p, x^p \rangle - c^p| \leq \max_{q \in [n] \backslash p} \mathrm{diam}(\mathcal{C}_q).$$

This also violates assumption (14). This implies that if (14) is satisfied, $H_p(Q)$ must separate $\mathcal{C}_p$ from $\cup_{q \neq p} \mathcal{C}_q$.

**Fact 2:** *The distance from any point in $\mathcal{C}_q$ from $H_p(Q)$ is bounded* as follows. For $x \in \mathcal{C}_q$, $q \neq p$, we have that

$$\mathcal{D}(x, H_p(Q)) \leq \mathcal{D}(x, x^q) + \mathcal{D}(x^q, H_p(Q)) \leq 2 \max_{q' \in [n] \backslash p} \text{diam}(\mathcal{C}_{q'}). \tag{59}$$

Now, we proceed to the main proof. For the ease of notation, we simply write $H_p$ for $H_p(Q)$.

**First**, from assumption (14), we has that for any $p \in [n]$,

$$\mathcal{D}(\mathcal{C}_p, H_p) = \min_{x \in \mathcal{C}_p} \mathcal{D}(x, H_p) = \min_{x \in \mathcal{C}_p} |c^p - \langle v^p, x \rangle|. \tag{60}$$

We have that

$$\min_{x \in \mathcal{C}_p} \mathcal{D}(x, H_p) > 2n \max_{q \neq p} \text{diam}(\mathcal{C}_q)$$

$$\geq \sum_{q \in [n] \backslash p} \max_{x \in \mathcal{C}_q} \mathcal{D}(x, H_p). \tag{61}$$

**Second**, we shows that how to pick a common point which exists when (61) is satisfied. Let us choose a collection of points $x^p \in \mathcal{C}_p$, $p \in [n]$, and define

$$x^\star = \frac{1}{n} \sum_{p \in [n]} x^p.$$

Now, we show that $x^\star \in E_p$, $\forall p \in [n]$.

For each $p \in [n]$, consider $H_p$. We denote

$$\zeta_{pp} := c^p - \langle v^p, x^p \rangle > 0;$$
$$\zeta_{pq} := \langle v^p, x^q \rangle - c^p > 0, \quad q \neq p.$$

Note that $\mathcal{D}(x, H_p) = |\langle v^p, x \rangle - c^p|$. Follows (61), we have that

$$\zeta_{pp} \geq \min_{x \in \mathcal{C}_p} \mathcal{D}(x, H_p) > \sum_{q \in [n] \backslash p} \max_{x \in \mathcal{C}_q} \mathcal{D}(x, H_p) \geq \sum_{q \in [n] \backslash p} \zeta_{pq}. \tag{62}$$

Now, let consider

$$\langle v^p, x^\star \rangle = \frac{1}{n} \sum_{q \in [n]} \langle v^p, x^q \rangle = \frac{1}{n} \sum_{q \in [n] \backslash p} (c^p + \zeta_{pq}) + \frac{1}{n}(c^p - \zeta_{pp})$$

$$= c^p + \frac{1}{n} \left( \sum_{q \in [n] \backslash p} \zeta_{pq} - \zeta_{pp} \right) < c^p. \tag{63}$$

Therefore, $x^\star \in E_p$. As it is true for all $E_p$, one has that

$$x^\star \in \bigcap_{p \in [n]} E_p. \tag{64}$$

Finally, by Lemma 13, we can conclude that $x^\star$ is a common point. $\quad\square$

## C   Sample Complexity Analysis

***Proof of Lemma 16.*** Denote $\Delta$ as the simplex corresponding to $M = [x^1, ..., x^n]$, $\Delta_i$ as the facet opposite the vertex $x^i$, and $h_i(\Delta)$ is the height of simplex w.r.t. the vertex $x^i$. Denote $\text{Vol}_k(C)$ as the $k$-dimensional content of $C \subset \mathbf{E}^{n-1}$, where $\dim(C) = k$. Using simple calculus, one has that

$$h_i(\Delta) = \frac{1}{n-1} \frac{\text{Vol}_{n-1}(\Delta)}{\text{Vol}_{n-2}(\Delta_i)}, \tag{65}$$

We also denote $\hat{\Delta}$ as the perturbed simplex corresponding to $M + R$ and $\hat{\Delta}_i$ as the facet opposite the to the perturbation of $x^i$.

we bound the height $h_i(\hat{\Delta})$ for all $i \in [n-1]$. For the height $h_n(\hat{\Delta})$, one can apply similar reasoning. Let define the coordinate matrix and the pertubation matrix w.r.t $x^n$ as follows

$$V := \text{coM}(M, n), \quad U := \text{coM}(R, n); \tag{66}$$

We have that $|U_{ij}| < \epsilon, \ \forall i, j$. By the definition of width $\vartheta(M)$, we have that

$$\sigma_{n-1}(V) \geq \vartheta(M) \geq \sigma. \tag{67}$$

Let define the Gram matrix and perturbed Gram matrix as follows

$$\begin{aligned} G &:= V^\top V \\ \hat{G} &:= (V + U)^\top (V + U). \end{aligned} \tag{68}$$

One has that,

$$\hat{G} - G = V^\top U + U^\top V + U^\top U := \overline{U}.$$

One has that $\overline{U}_{ij} \leq \overline{\epsilon} := 3n\epsilon$, as $|V_{ij}| < 1$ and $|U_{ij}| < \epsilon < 1$. We also has that $\|\overline{U}\|_2 \leq \|\overline{U}\|_F \leq n\overline{\epsilon}$.

**First step**. we bound the quantity $\frac{|\det(G+\overline{U}) - \det(G)|}{|\det(G)|}$. By [15]'s Corollary 2.14, one has that

$$\frac{|\det(G + \overline{U}) - \det(G)|}{|\det(G)|} \leq \left(1 + \frac{\|\overline{U}\|_2}{\sigma_{n-1}(G)}\right)^{n-1} - 1 \leq \left(1 + \frac{n\overline{\epsilon}}{\sigma^2}\right)^n - 1. \tag{69}$$

As $(1 + z)^n \leq \frac{1}{1 - nz}$ when $z \in (0, \frac{1}{n})$ and $n > 0$. One has that

$$\frac{|\det(G + \overline{U}) - \det(G)|}{|\det(G)|} \leq \frac{n^2\overline{\epsilon}}{\sigma^2 - n^2\overline{\epsilon}}, \tag{70}$$

when $\overline{\epsilon} \leq \frac{\sigma^2}{n^2}$, or $\epsilon \leq \frac{\sigma^2}{3n^3}$. Let us define $k := \frac{\sigma^2 - n^2\overline{\epsilon}}{n^2\overline{\epsilon}}$, one has that

$$\frac{|\det(G + \overline{U}) - \det(G)|}{|\det(G)|} \leq \frac{1}{k}. \tag{71}$$

It means that

$$\det(G + \overline{U}) \geq \left(1 - \frac{1}{k}\right)\det(G) \tag{72}$$

**Second step.** we bound the change in content of the $i^{\text{th}}$ facets of the simplex, for $i \in [n-1]$. Consider the facet that is opposite to the vertex $x^i$, and denote $V(i), U_i$ as the sub-matrices of $V, U$ by removing $i^{\text{th}}$ column. Denote the Gram matrix

$$\begin{aligned} G(i) &:= V(i)^\top V(i) \\ \hat{G}(i) &:= (V(i) + U(i))^\top (V(i) + U(i)) \end{aligned} \tag{73}$$

Note that one can obtain $G(i), \hat{G}(i)$ and by removing $i^{\text{th}}$ row and column of $G(i), \hat{G}(i)$ respectively. Denote $\overline{U}(i) := \hat{G}(i) - G(i)$, we has that all entries of $\overline{U}(i)$ smaller than $\overline{\epsilon}$.

Moreover, by Singular Value Interlacing Theorem, one has that

$$\sigma_1(G) \geq \sigma_1(G(i)) \geq \sigma_2(G) \geq \sigma_2(G(i)) \geq \cdots \geq \sigma_{n-2}(G(i)) \geq \sigma_{n-2}(G(i)) \geq \sigma_{n-1}(G). \tag{74}$$

Similarly, one has that

$$\frac{|\det(G(i) + \overline{U}(i)) - \det(G(i))|}{|\det(G(i))|} \leq \left(1 + \frac{\|\overline{U}(i)\|_2}{\sigma_{n-2}(G(i))}\right)^{n-2} - 1 \leq \left(1 + \frac{n\overline{\epsilon}}{\sigma^2}\right)^n - 1 \leq \frac{1}{k}. \tag{75}$$

It means that
$$\det(G(i) + \overline{U}(i)) \le \left(1 + \frac{1}{k}\right)\det(G(i)). \tag{76}$$

**Third step.** We bound the height $h_i$ corresponding to the vertices $x^i$ in this step. For $i \in [n-1]$ one has that
$$\mathrm{Vol}_d(\hat{\Delta}) = \frac{1}{(n-1)!}\sqrt{\det(G+\overline{U})}.$$
$$\mathrm{Vol}_{d-1}(\hat{\Delta}_i) = \frac{1}{(n-2)!}\sqrt{\det(G(i)+\overline{U}(i))}. \tag{77}$$

Furthermore, by the Eigenvalue Interlacing Theorem, we have $\det(G)/\det(G_i) \ge \sigma_{n-1}(G) \ge \sigma^2$. Putting things together, one has that

$$h_i(\hat{\Delta}) = \frac{1}{n-1}\frac{\mathrm{Vol}_{n-1}(\hat{\Delta})}{\mathrm{Vol}_{n-2}(\hat{\Delta}_i)} = \sqrt{\frac{\det(G+\overline{U})}{\det(G(i)+\overline{U}(i))}} \ge \sqrt{\frac{k-1}{k+1}\frac{\det(G)}{\det(G(i))}} \ge \sigma\sqrt{\frac{\sigma^2 - 6n^3\epsilon}{\sigma^2}} \tag{78}$$

We note that, the above holds true for $i \in [n-1]$.

**Fourth Step.** Now, we bound the height corresponding to the vertex $x^n$. We can define the coordination matrix and pertubation matrix w.r.t $x^1$ as follows.
$$V' = \mathrm{coM}(M,1), \quad U' = \mathrm{coM}(R,1). \tag{79}$$

Note that, by the definition of the width, we have that
$$\sigma_{n-1}(V') \ge \vartheta(M) \ge \sigma; \tag{80}$$
and also, $|U'_{ij}| \le \epsilon$. Similarly, applying Steps 1-3, we also have that
$$h_n(\hat{\Delta}) \ge \sqrt{\sigma^2 - 6n^3\epsilon}$$

Therefore, $h_i(\hat{\Delta}) \ge \sqrt{\sigma^2 - 6n^3\epsilon}$ holds true for all $i \in [n]$. We conclude that
$$h_{\min} \ge \sqrt{\sigma^2 - 6n^3\epsilon}, \tag{81}$$
whenever, $\epsilon \le \frac{\sigma^2}{3n^3}$. $\qquad\square$

***Proof of Theorem 18.*** Note that $|\mathcal{P}| = n$. Denote $\epsilon_0 := 2\max_{p\in[n]}\mathrm{diam}(\mathcal{C}_p)$. For any $\omega_p \in \mathcal{P}$, define the event
$$\mathcal{E} = \{\phi^{\omega_p} \in \mathcal{C}_p, \forall p \in [n]\}$$
By the construction of the confidence set, we guarantee that $\mathcal{E}$ happen with probability at least $1 - n^2\delta$.

Consider $p \in [n]$, for any $q \in [n] \setminus p$, let $x^q$ be the projection of $\phi^{\omega_q}$ onto $H_p$, and $x^p := \arg\min_{p\in\mathcal{C}_p}\mathcal{D}(x, H_p)$. We have that
$$\mathcal{D}(x^k, \phi^{\omega_k}) \le \epsilon_0, \ \forall k \in [n].$$

We need to bound $\mathcal{D}(x^p, H_p)$ by bounding the minimum height of simplex $\mathrm{Conv}\left(\{x^p\}_{p\in[n]}\right)$, which is a pertubation of $\mathrm{Conv}\left(\{\phi^{\omega_p}\}_{p\in[n]}\right)$.

Define matrix $M = [\phi^{\omega_p}]_{p\in[n]}$, and $\hat{M} = [x^p]_{p\in[n]}$. Let $R := M - \hat{M}$ be the perturbation matrix, one has that $R_{ij} \le \epsilon_0, \ \forall(i,j)$. By Lemma 16, we have that
$$\mathcal{D}(x^p, H_p) \ge \sqrt{\sigma^2 - 12n^3\epsilon_0} \tag{82}$$

Therefore, for $\mathcal{D}(x^p, H_p) \ge n\epsilon_0$ holds , it is sufficient to provide the condition for $\sigma$ such that
$$\sqrt{\sigma^2 - 12n^3\epsilon_0} \ge n\epsilon_0. \tag{83}$$

Assuming that $\epsilon_0 < 1$, for the condition of Lemma 16 and the above inequality to hold, it is sufficient to choose

$$\epsilon_0 = \frac{\sigma^2}{13n^3}.$$

Now, we calculate the upper bound for sample needed. At epoch $K$, we have that

$$\epsilon_0 = 2\mathrm{diam}(\mathcal{C}_p) \geq 4\sqrt{\frac{2n\log(\delta^{-1})}{K}}$$
$$\sigma = \frac{n\varsigma}{c_W}. \tag{84}$$

Then we have $K = O\left(\frac{n^{13}\log(n\delta^{-1}\varsigma^{-1})}{\varsigma^4}\right)$. As each phase, there are at most $n^2$ queries, then the total number of sample needed is

$$T = O\left(\frac{n^{15}\log(n\delta^{-1}\varsigma^{-1})}{\varsigma^4}\right) \tag{85}$$

for the algorithm to return a common point, with probability of at least $1 - \delta$. $\square$

***Proof of Theorem 19.*** The proof is identical to that of Theorem 18, with the width of the simplex bounded by $\vartheta(W) \geq \frac{n\varsigma}{c_W}$. $\square$

## D    Examples of Strictly Convex Games

Consider a facility-sharing game (a generalisation of cost-sharing games [2, 1]) where joining a coalition $S$ would provide each player of that coalition a utility value $v(k)$ where $|S| = k$, and they have to pay an average maintenance cost $c(k)$. The expected reward of $S$ is defined as $\mu(S) = v(k) - c(k)$, representing the average utility of its coalitional members. This setting represents many real-world scenarios, such as:

- University departments together plan to set up and maintain a shared computing lab. The value of using the lab is the same $v(k) = v$ for each department (e.g., their students can have access computing facilities), but the maintenance cost $c(k)$ is monotone decreasing and strictly concave (e.g., the more participate the less the average maintenance cost becomes). An example to such maintenance cost function is, e.g., $c(k) = C_1 - C_2 k^\alpha$, where $\alpha > 1$ and $C_1, C_2$ are appropriately set constants such that the total maintenance cost $kc(k) = C_1 k - C_2 k^{(\alpha+1)}$ is non-negative and monotone increasing in the $[0, n]$ range ($n$ is the total number of departments). A coalition $S$ here represents the cooperating departments.

- An international corporate is expanding its international markets via mutual collaborations with multiple local companies. The cost for each potential local member company to join the consortium is fixed $c$ (e.g., integration cost), while the benefit they can gain through this collaboration, $v(k)$, is a strictly monotone convex function in the number of local markets $k$ (assuming that each local partner is in charge of their own local market), due to the potential synergies between different markets. An example benefit/utility function is, e.g., $v(k) = k^\alpha$ with $\alpha > 1$. The corporate's task is then to invite local companies to join their coalition/consortium $S$.

- (An alternative version of airport games) Airlines decide whether they launch a flight from a particular airport. The more airlines decide to do so, the higher value $v(k)$ (e.g., more connection options), and the lower average buy-in cost $c(k)$ (e.g., runway maintenance, staff cost etc.) each airlines can have. It's reasonable to assume that $v(k)$ is strictly convex and monotone increasing (e.g., the number of connecting combinations grows exponentially) and $c(k)$ is monotone decreasing and strictly concave. Those airlines who decide to invest into that airport will form a coalition.

For each of the scenarios above, we can see that the expected reward function $\mu(S) = v(|S|) - c(|S|)$ is indeed strictly supermodular. The reason is that $v(k)$ and $c(k)$ are discrete and finite on $[0, n]$, and thus, we can easily find $\varsigma > 0$ for which these games also admit $\varsigma$-strict convexity.

# E   Further Discussions

## E.1   Comparison with Pantazis *et al.* [20]

While the algorithm in [20] is proposed for general cooperative games and conceptually applicable to the class of strictly convex games, we argue that their algorithm is not statistically and computationally efficient when applied to strictly convex games, due to the absence of a specific mechanism to leverage the supermodular structure of the expected reward function. In particular, firstly, we argue that without any modification and with bandit feedback, their algorithm would require a minimum of $\Omega(2^n)$ samples. Secondly, although we believe the framework of [20] could be conceptually applied to strict convex games, significant non-trivial modifications may be necessary to leverage the supermodular structure of the mean reward function.

**Appplying [20] to strictly convex games without any modifications.**   We first briefly outline their algorithmic framework. In this paper, the authors assume that each coalition $S \subset N$ has access to a number of samples, denoted as $t_S > 1$. For each coalition $S$, the empirical mean is denoted as $\overline{\mu}_{t_S}(S)$, and a confidence set for the given mean reward is constructed, denoted as,

$$\mathcal{C}(\mu(S)) = \left\{ \hat{\mu}(S) \in [0,\ 1] \mid |\hat{\mu}(S) - \overline{\mu}_{t_S}(S)| \leq \varepsilon_{t_S} \right\}, \text{ for some } \varepsilon_{t_S} > 0 \,.$$

We note that while the algorithm in [20] constructs the confidence set using Wasserstein distance, in the case of distributions with bounded support, we can simplify it by using the mean reward difference. After constructing the confidence set for the mean reward of each coalition, the algorithm solves the following robust optimization problem:

$$\min_{x \in \mathbb{R}^n}\ \|x\|_2^2$$
$$\text{s.t. } x(N) = \mu(N)$$
$$x(S) \geq \sup(\mathcal{C}(\mu(S)), \quad \forall S \subset N.$$

That is, finding the stable allocation for the worst-case scenario within the confidence sets. It is clear that when directly applying this framework to the bandit setting, each coalition must be queried at least once, that is $t_S > 1$. This inevitably leads to a complexity of $\Omega(2^n)$ samples, regardless of the sampling scheme one employs. In term of computation, with $2^n - 2$ confidence sets for all coalitions $S \subset N$, tabular representation of the confidence set incurs extreme computational cost.

**Significant modifications required for [20].**   As described above, the algorithm in [20] suffers from $2^n$ sample complexity, and the main reason is because it requires constructing confidence sets for the mean reward for all coalitions $S \subset N$. As such, if we want to apply their algorithm efficiently to the bandit setting, we need to address this limitation.

To do so, one may need to develop an approach to design a confidence set for a specific class of strictly convex games. For instance, we can consider the following approach: Given historical data, instead of writing a confidence set for each individual coalition, let us define a confidence set for the mean reward function as follows:

$$\mathcal{C}(\mu) = \left\{ \hat{\mu} : 2^N \to [0,\ 1] \mid \hat{\mu} \in [\mathcal{C}(\mu(S))]_{S \subset N},\ \hat{\mu} \text{ is strictly supermodular} \right\}; \tag{86}$$

where the confidence set $\mathcal{C}(\mu(S))$ could potentially be $[0, 1]$ for some coalition $S$, as there is no data available for these coalitions. Let $\text{Core}(\hat{\mu})$ be the core with respect to the reward function $\hat{\mu}$. We propose a generalization of the framework from the robust optimization problem to adapt to the structure of the game as follows.

$$\min_{x \in \mathbb{R}^n}\ \|x\|_2^2$$
$$\text{s.t. } x(N) = \mu(N) \tag{87}$$
$$x \in \bigcap_{\hat{\mu} \in \mathcal{C}(\mu)} \text{Core}(\hat{\mu}).$$

That is, we find a stable allocation $x$ for every possible supermodular function within the confidence set of the reward function.

However, implementing and analyzing this approach may pose significant challenges. The first challenge lies in constructing a tight confidence set $[\mathcal{C}(\mu(S))]_{S \subset N}$ such that all functions within

this collection are strictly supermodular. We are not aware of a method to explicitly construct $[\mathcal{C}(\mu(S))]_{S \subset N}$ containing only strictly supermodular functions, and we believe this set could potentially be very complicated. To illustrate, consider the scenario where we have samples from two coalitions, $\{1\}$ and $\{1, 2\}$, with the following empirical means:

$$\overline{\mu}(\{1\}) = 0.11; \quad \overline{\mu}(\{1, 2\}) = 0.1$$

This situation might occurs when the number of samples is insufficient. In such cases, regardless of the value chosen for the remaining coalition rewards in the function $\overline{\mu}(S)$, $\overline{\mu}(S)$ is not supermodular (as $\{1\} \subset \{1, 2\}$, yet $\overline{\mu}(1) > \overline{\mu}(1, 2)$). Consequently, either the confidence set $\mathcal{C}(\mu(1))$ or $\mathcal{C}(\mu(1, 2))$ does not contain the empirical mean reward, indicating the highly complicated shape of the confidence set.

The second challenge is that while computing a stable allocation for a given supermodular reward function $\hat{\mu}$ is a straightforward task, computing a stable allocation for all supermodular reward functions in the confidence set $\mathcal{C}(\mu)$ in a computationally efficient way is an open problem, to the best of our knowledge.

The discussion above also highlights the key difference between our work and that of [20]: Instead of explicitly constructing the confidence set of the expected mean reward function to integrate the supermodular structure for computing a stable allocation, which might be a sophisticated task, we directly exploit the geometry of the core of strictly convex games. Specifically, in strictly convex games, each vertex of the core corresponds to a marginal vector with respect to some permutation orders. Given that one can construct the confidence set of marginal vectors easily, our method is conceptually and computationally simpler. However, we believe that adopting the more general framework of robust optimization as presented in [20] is a very interesting, but non-trivial, direction, and we leave it for future work.

## E.2 Comment on Lower Bounds

It is also crucial to develop the lower bound for the class of strict convex game. One promising direction is to extend the game instances in Theorem 7. However, there could be several technical issues when it comes to deriving a meaningful lower bound for strictly convex games. The main problem is twofold: Firstly, not every small perturbation of the face game may result in a strictly convex game; therefore, careful tailoring to ensure strict convexity is required. Second, to show a polynomial dependence of sample complexity on $n$, we need to generalise the two game instances in Theorem 7 into $\text{poly}(n)$ game instances for the information-theoretic argument. It is not clear how to construct them so that their core has no intersection, and the statistical distance of the reward can be upper bounded. This can be further detailed as follows:

Firstly, as proven in [14], a face game is only guaranteed to be a convex game, and not all perturbations of it can result in a strictly convex game. In fact, we believe great care is required to ensure that the perturbations of the face games are strictly convex, thereby allowing them to be used to derive lower bounds for strictly convex games. Moreover, even if one can guarantee that the perturbation of the two face-game instances in Theorem 7 is strictly convex, they can only result in a very loose lower bound. Particularly, The two game instances in Theorem 7 are originally constructed using two strictly convex games $G_0$ and $G_1$, whose expected rewards differ in *only one coalition*, denoted as $C \subset N$. This setup simplifies the computation of the statistical distance of the face-games which are perturbations of $G_0$ and $G_1$ *corresponding to the same coalition* $C$. However, since only two game instances are used to derive the result of Theorem 7, if we employ fully-dimensional-core perturbed game instances of them, the resulting lower bound will be independent of the dimension $n$. In other words, the finite-sample lower bound can be very loose and does not show any dependence on the dimension $n$.

Secondly, generalising the approach to $\text{poly}(n)$ game instances is not straightforward, as it requires choosing $\text{poly}(n)$ coalitions to perturb the rewards of the original game $G_0$, not just one coalition. This results in significant differences in the expected reward functions of the corresponding face games, as each face game is a perturbation with respect to several different coalitions. Consequently, upper-bounding the pairwise KL distance between these games is highly nontrivial and would require sophisticated exploitation of the structure of strictly convex games.

