# OpenReview forum: "Learning the Expected Core of Strictly Convex Stochastic Cooperative Games"
_NeurIPS.cc/2024/Conference — NeurIPS 2024 poster_

### Official Review · Reviewer_f4nV · 2024-07-12

**Soundness:** 3
**Presentation:** 3
**Contribution:** 3
**Rating:** 6
**Confidence:** 3

**Summary:**

The paper tries to find an expected core under the assumption that a characteristic function is $\varsigma$-strictly convex cooperation game.
The authors provide a bandit-based sampling algorithm called Common-Points-Picking, which allows us to compute the expected core in a polynomial number of samples.
They prove that strictly convex is a sufficient condition to find an expected core with the polynomial number of samples.

**Strengths:**

s1. The paper discusses the learnability of the expected core and shows that strict convexity is a sufficient condition to guarantee thelearnability.

s2. The paper also provides a novel algorithm based on convex geometry and the proposed algorithm outputs an expected core with probability 1 - $\delta$.

s3. The sample complexity of the proposed algorithm is polynomial in the size of the players.

**Weaknesses:**

w1. My main concern is that the assumption of strict convexity is not very natural while providing the hardness of learnability in a non-strict convexity situation is good.
It would be more better to show the validity of the assumption by showing that some known application of cooperative games satisfies it. For example, it is better to provide a discussion that in some known convex games, such as induced subgraph games with positive weights[1], airport games[2], and some others[3-6], which parameters of these convex games are related to the parameter \varsigma.

w2. Providing the upper bound of sampling complexity is good, but I cannot measure how tight the bound is. Since the paper provides a discussion of the hardness of calculating the lower bound, providing the tight lower bound of sample complexity is ideal.

w3. It would be better to discuss other sampling-based algorithms for computing core. For example, [7] provides an FPRAS algorithm for the convex(supermodular) cooperative game, and [8] provides a PAC-learning-based algorithm for finding the core. While they each have different assumptions and problem settings, the discussion about them is useful for readers to clarify where the paper stands.

[1] Deng, X., & Papadimitriou, C. H. (1994). On the complexity of cooperative solution concepts. Mathematics of operations research, 19(2), 257-266.

[2] Littlechild, S. C., and Owen, G. 1973. A simple expression for the shapley value in a special case. Management Science 20(3):370–372.

[3] Oishi, T., & Nakayama, M. (2009). Anti-dual of economic coalitional TU games. The Japanese Economic Review, 60, 560-566.

[4] Graham, D. A., Marshall, R. C., & Richard, J. F. (1990). Differential payments within a bidder coalition and the Shapley value. The American Economic Review, 493-510.

[5] Feigenbaum, J.; Papadimitriou, C. H.; and Shenker, S. 2001. Sharing the cost of multicast transmissions. Journal of computer and system sciences 63(1):21–41.

[6] O'Neill, B. (1982). A problem of rights arbitration from the Talmud. Mathematical social sciences, 2(4), 345-371.

[7] Liben-Nowell, David, et al. "Computing shapley value in supermodular coalitional games." Computing and Combinatorics: 18th Annual International Conference, COCOON 2012, 2012.

[8] Igarashi, A., Sliwinski, J., & Zick, Y. (2019). Forming probably stable communities with limited interactions. In Proceedings of the AAAI Conference on Artificial Intelligence (Vol. 33, No. 01, pp. 2053-2060).

**Questions:**

Could you please comment about the weakness comments if I have misunderstood something? The following is a comment rather than a question.

q1. Is there any relationship between totally balanced games [9]? Totally balanced condition is a necessary and sufficient condition for guaranteeing the existence of the core. It may be useful for generalizing the proposed problem setting.

[9] Shapley, L. S., & Shubik, M. (1969). On the core of an economic system with externalities. The American Economic Review, 59(4), 678-684.

**Limitations:**

Yes, they adequately address the limitations.

---

> ### Author Rebuttal · Authors · 2024-08-06
>
> # Response to reviewer f4nV
> We thank the reviewer for the insightful comments. Below are our responses/clarifications to your questions:
>
> ## Comment 1: Practical example is needed.
> ### Response:
>
> Consider a facility-sharing game (a generalisation of cost-sharing games [1, 2]) where joining a coalition $S$ would provide each player of that coalition a utility value $v(k)$ where $|S| = k$, and they have to pay an average maintenance cost $c(k)$. The expected reward of $S$ is defined as $\mu(S) = v(k) - c(k)$, representing the average utility of its coalitional members.
> This setting represents many real-world scenarios, such as:
>
> - University departments together plan to set up and maintain a shared computing lab. The value of using the lab is the same $v(k) = v$ for each department (e.g., their students can have access computing facilities), but the maintenance cost $c(k)$ is monotone decreasing and strictly concave (e.g., the more participate the less the average maintenance cost becomes). An example to such maintenance cost function is, e.g., $c(k) = C_1 - C_2k^\alpha$, where $\alpha > 1$ and $C_1, C_2$ are appropriately set constants such that the total maintenance cost $kc(k) = C_1k - C_2k^{(\alpha +1)}$ is non-negative and monotone increasing in the $[0,n]$ range ($n$ is the total number of departments).
>
> - (An alternative version of airport games) Airlines decide whether they launch a flight from a particular airport. The more airlines decide to do so, the higher value $v(k) $ (e.g., more connection options), and the lower average buy-in cost $c(k)$ (e.g., runway maintenance, staff cost etc.) each airlines can have. It's reasonable to assume that $v(k)$ is strictly convex and monotone increasing (e.g., the number of connecting combinations grows exponentially) and $c(k)$ is monotone decreasing and strictly concave. An example for strictly convex and monotone increasing benefit function is, e.g., $v(k) = k^\alpha$ with $\alpha > 1$.
>
> For each of the scenarios above, we can see that the expected reward function $\mu(S) = v(|S|) - c(|S|)$ is indeed strictly supermodular.
> In addition, due to the fact that $v(k)$ and $c(k)$ are discrete and finite on $[0,n]$, we can easily find $\varsigma > 0$ for which these games also admit $\varsigma$-strict convexity.
>
> ## Comment 2: Discussion on parameters of practical convex games and the conditions for strict convexity.
> ### Response:
> It is indeed challenging to determine the conditions under which the core of popular games, such as induced subgraph games and airport games, is full-dimensional, let alone strictly convex. We conjecture that additional conditions on the parameters of the game would be needed to guarantee either strict convexity or full-dimensionality of the core, which would enable the learnability of the problem.
>
> While the primary goal of this paper is to identify a general sufficient condition to achieve polynomial sample complexity, it is indeed interesting to investigate the conditions for particular games. Therefore, we leave this to future work.
>
> Note that in our response to your Comment 1 (see above), we have demonstrated how strict convexity can occur in other types of popular games such as cost/facility sharing games, when additional structures of the reward functions are included.
>
> ## Comment 3:  It would be better to discuss other sampling-based algorithms for computing core.
> ### Response:
> We thank the reviewer for the suggestion of comparing our result to the literature on Shapley value approximation and learning PAC-stable allocations from samples.
>
> Compared to [3] and other works on Shapley approximation, their key limitation is that they can only return a value that is within a bounded distance of the Shapley value, which may not necessarily be in the core. Our algorithm is designed to directly return a point in the core instead.
>
> Regarding the literature on learning PAC-stable allocations from given samples [4, 5], the goal is to find an $(\delta, \epsilon)$-PAC stable allocation $x$, that is, an allocation that satisfies
> $$
> \mathrm{Pr}[(1-\epsilon) x(S) \leq f(S)] \geq 1-\delta.
> $$
> In some sense, our result can be understood as $(\delta, 0)$-PAC. However, to achieve $\epsilon = 0$, we require active sampling, not just from a given set of data, and also assumptions on the strict convexity of the game. As such, existing PAC-stable allocations would not work well in our setting.
>
> ## Comment 4: Relation to balanced game.
> ### Response:
> We thank the reviewer for the suggestion. Although a totally balanced game is both a necessary and sufficient condition for guaranteeing the non-emptiness of the core, it is unclear how to strengthen the condition of a totally balanced game to guarantee the full-dimensionality of the core, which enables the learnability of the problem. The setting of totally balanced games and its further conditions is indeed an interesting area that we wish to explore in future work.
>
> ## Comment 5: on the lower bound.
> ### Response:
>
> We agree that having a matching lower bound would be ideal. As currently we do not have it, we have conducted some additional experiments to compare the sample complexity our algorithm needs in practice with the derived theoretical upper bound from Theorem 19. Due to space constraints, we refer to the discussion with Reviewer 1Z68 (Comment 1) for the simulation details. The simulation results can be found in the attached PDF and they show that our theoretical upper bound is comparable with the empirical numbers (Figure 1); that is, the order of the polynomial in $n$ more or less matches the experimental results.
>
> ### References:
>
> [1]Aadland\&Kolpin. Shared irrigation costs: an empirical and axiomatic analysis. 1998.
>
> [2]Ambec\&Ehlers. Sharing a river among satiable agents. 2008.
>
> [3]Liben-Nowell et al. Computing shapley value in supermodular coalitional games. 2012.
>
> [4]Balcan et al. Learning cooperative games, 2015.
>
> [5]Balkanski et al. Statistical cost sharing. 2017.

---

> > ### Author Response · Authors · 2024-08-12
> >
> > Dear Reviewer,
> >
> > Thank you for your time and effort in reviewing our paper. We hope our responses have addressed your concerns and questions. If you have any further questions, please don’t hesitate to let us know.
> >
> > Best regards,
> >
> > The Authors

---

> > > ### Comment · Reviewer_f4nV · 2024-08-14
> > >
> > > Thanks for taking the time to address my questions. I have no further question.
> > >
> > > With respect to the assumption that I raised as a main concern, It would be good to add an example the authors wrote in the rebuttal because the convexity assumption is one of the major assumptions in the cooperative game community, but not so much in the machine learning community.

---

> > > > ### Author Response · Authors · 2024-08-14
> > > >
> > > > Thank you very much for your suggestion. We will incorporate the examples we wrote in the rebuttal phase into the final version of the paper.
> > > >
> > > > Best regards,
> > > > The Authors

---

### Official Review · Reviewer_oJVQ · 2024-07-12

**Soundness:** 3
**Presentation:** 3
**Contribution:** 3
**Rating:** 5
**Confidence:** 3

**Summary:**

This paper studies the problem of learning the expected core when only bandit feedback is available, under the assumption that the problem is strictly convex. They proposed Common-Point-Picking (CPP) algorithm that returns a point in the expected core given an oracle that provides noisy samples of the unknown (full-dimensional) simplex's vertices. Sample complexity analysis is provided.

**Strengths:**

The paper is in general well-written and smooth to follow. The technical proofs seem to be rigorous. The proposed CPP algorithm is novel to me and has a sound theoretical guarantee.

**Weaknesses:**

Although the paper is basically theoretical, I would appreciate it if simulation results could be provided. For example, the authors could plot the actual number of samples used to reach the given precision to validate the sample complexity results. Also, as this paper is motivating CPP using geometric intuitions, some illustrations about this would be helpful. Currently, the only simulation result is providing empirical evidence of the conjecture that $C_W$ is relatively small. Moreover, although full-dimensionality is used as an intuition for assuming strict convexity, the latter seems too strong.

**Questions:**

- Figure 2 shows that $C_W$ tends to be long-tailed, which makes the empirical validation relatively weak. Could the authors comment more on this?
- If the strict convexity assumption is replaced with purely the full dimensionality assumption, do the authors expect CPP to work?

**Limitations:**

I don't see any limitations or potential negative societal impact of this work.

---

> ### Author Rebuttal · Authors · 2024-08-06
>
> # Responses to Reviewer oJVQ
>
> We thank the reviewer for the insightful comments. Below are our responses/clarifications to your questions:
>
> ## Comment 1 - Simulation results for sample complexity
> ### Response:
> To illustrate the sample complexity of our algorithm in practice and how it is compared to our theoretical upper bound, we have conducted a simulation as described below.
>
> **Simulation setting:**
> We generate convex game of $n$ players with the expected reward function $f$ defined recursively as follows:
> For each $S \subset N$,
> $$
> f(S \cup \{i\}) = f(S) + |S| + 1 + 0.9\omega,
> $$
> for some $\omega$ sampled i.i.d. from the uniform distribution $\mathrm{Unif}([0,1])$.
> We then normalize the value of the reward function within the range $[0,1]$.
> The strict convexity constant is $\varsigma \approx 0.1/n$.
> We plot the samples required by the algorithm to find a point in the true core in the attached PDF file (Figure 1).
>
> From the simulation results, we can see that the growth pattern nearly matches that of the theoretical bound given in Theorem 19 in our paper, indicating that our theoretical bound is highly informative.
>
> ## Question 1: If the strict convexity assumption is replaced with purely the full dimensionality assumption, do the authors expect CPP to work?
> ### Response:
> Our algorithm exploits the property of convex games where each vertex corresponds to some marginal vectors. Hence, convexity is necessary. However, we expect that the CPP algorithm can work well even when the strict convexity assumption is violated and replaced by convexity and full-dimensionality.
> In fact, our algorithm operates quite independently of the strict convexity assumption, as it does not require the knowledge of the strict convexity constant $\varsigma$ neither the width constant $c_W$ of the function.
>
> In more detail, from a theoretical perspective, strict convexity allows us to provide a provable approach to find an input for the algorithm easily, which is the set of any permutation and its adjacent permutations.
> However, in practice, the algorithm works independently of the strict convexity assumption by using the collection of cyclic permutations $\mathfrak{C}_n$ as the input.
>
> To demonstrate that our algorithm is indeed still robust even when the strict convexity assumption is violated, we ran a simulation where the characteristic function is only convex, or the strict convexity constant is arbitrarily small as follows:
>
> **Simulation setting:**
> We generate convex game of $n$ player with the expected reward function $f$ defined recursively as follows:
> For each $S \subset N$,
> $$
> f(S \cup \{i\}) = f(S) + |S| + 1 + \omega.
> $$
> for some $\omega$ sampled i.i.d. from the uniform distribution $\mathrm{Unif}([0,1])$.
> We then normalize the value of the reward function within the range $[0,1]$.
>
> To see that the strict convexity constant $\varsigma$ can be $0$, consider the example:
> Let $S = \\{1\\}$, and $T=\\{1,2\\}$, and suppose that
> $$
> f(S\cup \{3\}) = f(S) + 3, \text{suppose that $\omega = 1$}; \quad \quad  f(T\cup \{3\}) = f(T) + 3,  \text{suppose that $\omega = 0$}.
> $$
> Therefore, the marginal contribution of player $3$ to both $S$ and $T$ is $3$, hence the game is only convex, not strictly convex.
>
> We then generate stochastic rewards following the Bernoulli distribution $r_t(S) \sim \mathrm{Ber}(f(S))$.
> For each $n \in \\{2,...,10\\}$ we ran $100$ game samples.
> We use the cyclic permutations $\mathfrak{C}_n$ as the input for the algorithm.
> We plot the samples required by the algorithm to find a point in the true core in the attached PDF file (Figure 2).
> On the log scale, one can see that the number of samples required as $n$ grows is sub-exponential, indicating that our algorithm is robust when the strict convexity assumption is violated.
>
> ## Question 2: Figure 2 shows that tends to be long-tailed, which makes the empirical validation relatively weak. Could the authors comment more on this?
> ### Response:
> From our simulation, we can observe that with significantly high probability (i.e., 99.6 percent), the constant $c_W$ falls into the $[0,30]$ interval, independently from other parameter settings. This indicates that this constant is relatively small for the majority of strictly convex game instances.

---

> > ### Author Response · Authors · 2024-08-12
> >
> > Dear Reviewer,
> >
> > Thank you for your time and effort in reviewing our paper. We hope our responses have addressed your concerns and questions. If you have any further questions, please don’t hesitate to let us know.
> >
> > Best regards,
> > The Authors

---

### Official Review · Reviewer_1Z68 · 2024-07-12

**Soundness:** 3
**Presentation:** 4
**Contribution:** 3
**Rating:** 7
**Confidence:** 3

**Summary:**

The paper studies the problem of finding the core for Reward allocation
when the information about reward functions is incomplete.
Specifically, previous works either study deterministic
games and assume that the reward function is known, or study stochastic games
and assume that the reward distribution is known.
In contrast, this paper assumes that we have access to an oracle
that takes as input a coalition and outputs a stochastic reward for that coalition.
The paper obtains an algorithm for the special case of strictly convex games
which outputs a point in the expected core and with high probability uses as most
a polynomial number of samples.

The main idea behind the algorithm is as follows. If there was no issue with the noise,
then one could simply take the marginal vector of an arbitrary distribution as done by [21].
The noise prohibits us from estimating this vector exactly however and one can form only a confidence set.
The main idea is to form multiple confidence sets around multiple marginal vectors.
Next, we look for *common points* which are points $p$ with the following property:
if we choose one arbitrary point from each of the confidence sets, and calculate
the convex combination of these points, then this will contain $p$.
Perhaps surprisingly, this set is non-empty.

**Strengths:**

- The paper studies an interesting problem, and provides an elegant solution for it.
- The paper is well-written; the algorithm in particular is explained very well and is easy to understand.

**Weaknesses:**

- It would be good to obtain a lower bound.
  As is, the sample complexity is a (large) polynomial in $n$.
  Not clear to what extent this can be improved.
  While the authors discuss this briefly, it is an important drawback of the work.

**Questions:**

Questions:
- Are there any works that are similar to yours in terms of techniques?
  Seems to me like similar ideas should appear in the bandits literature but the only citation
  I can find is [14].


Typo: Equation 2 should require $C \ne S$.

---

> ### Author Rebuttal · Authors · 2024-08-06
>
> # Response to reviewer 1Z68
> We thank the reviewer for the insightful comments. Below are our responses/clarifications to your questions:
>
> ## Question 1: Are there any works that are similar to yours in terms of techniques?
> ### Response:
> Learning the core via sampling is typically considered to be a difficult problem within both the algorithmic game theory and learning theory communities, and thus, not many results have been published to date. Nevertheless, you are correct that our technique borrow some ideas from the bandit and learning theory literature:
>
> Our problem formulation can be viewed as finding a point within an unknown feasible set, defined by $f: 2^N\rightarrow [0,\;1]$.
> From this perspective, it is somewhat related to the literature on linear bandits with constraints [1, 2].
> However, existing techniques require the number of samples to scale with the number of free parameters that define the feasible set, which is not desirable in our problem as it leads to $\Omega(2^n)$ number of samples required.
> In contrast, we exploit the property of supermodular functions, where each marginal vector corresponds to a vertex of the feasible set, i.e., the core. Hence, we only need to estimate $n$ vertices.
> Moreover, we believe our CPP framework and *the development of an extension of the separating hyperplane is the first of its kind*, and we hope that it can find its application in other domains such as stochastic optimization and bandit theory.
>
> Regarding the impossibility results on sample complexity, the information-theoretic framework is typically used to derive lower bounds for bandit algorithms. While this framework is general, the main challenge and novelty of new lower bounds often lie in finding a collection of hard problem instances that satisfy the assumptions of the particular setting, which has led to various lower bounds in the literature [3, 4, 5].
> Our impossibility result follows this approach as the key challenge in deriving our Theorem 7 lies in constructing hard game instances such that their cores do not intersect, while the KL distance between the games is arbitrarily small, making it difficult to distinguish them with a finite number of samples.
> While existing techniques in the online learning literature are typically not suitable to derive lower bounds for our setting, we found that face-game problem instances [6] perfectly strike that balance, allowing us to derive the impossibility result through the information-theoretic framework.
>
> ## Comment 1: It would be good to obtain a lower bound. As is, the sample complexity is a (large) polynomial in $n$. Not clear to what extent this can be improved. While the authors discuss this briefly, it is an important drawback of the work.
>
> ### Response:
> We agree that having a matching lower bound would be ideal. As currently we do not have it, we have conducted some additional experiments to compare the sample complexity our algorithm needs in practice with the derived theoretical upper bound from Theorem 19.
>
> **Simulation setting:**
> We generate convex game of $n$ players with the expected reward function $f$ defined recursively as follows:
> For each $S \subset N$,
> $$
> f(S \cup \{i\}) = f(S) + |S| + 1 + 0.9\omega,
> $$
> for some $\omega$ sampled i.i.d. from the uniform distribution $\mathrm{Unif}([0,1])$.
> We then normalize the value of the reward function within the range $[0,1]$.
> The strict convexity constant is $\varsigma \approx 0.1/n$.
> We plot the samples required by the algorithm to find a point in the true core in the attached PDF file (Figure 1).
>
> From the simulation results, we can see that the growth pattern nearly matches that of the theoretical bound given in Theorem 19 in our paper, indicating that our theoretical bound is highly informative.
>
>
> ## References:
> [1] Shipra Agrawal and Nikhil R. Devanur. Bandits with concave rewards and convex knapsacks. Proceedings of the 15th ACM Conference on Economics and Computation, 2014.
>
> [2] Sanae Amani, Mahnoosh Alizadeh, and Christos Thrampoulidis. Linear stochastic bandits under safety constraints. In Advances in Neural Information Processing Systems, 2019.
>
> [3] Peter Auer, Nicolo Cesa-Bianchi, Yoav Freund, and Robert E. Schapire. The non-stochastic multi-armed bandit problem. SIAM journal on computing 32(1), 2002.
>
> [4] Paat Rusmevichientong and John N. Tsitsiklis. Linearly parameterized bandits. Mathematics of Operations Research, 35, 2010.
>
> [5] Robert Kleinberg, Aleksandrs Slivkins, and Eli Upfal. Bandits and experts in metric spaces. Journal of the ACM, 66, 2019.
>
> [6] Miguel Ángel Mirás Calvo, Carmen Quinteiro Sandomingo, and Estela Sánchez Rodríguez. The boundary of the core of a balanced game: face games. International Journal of Game Theory, 49, 2020

---

> > ### Comment · Reviewer_1Z68 · 2024-08-11
> >
> > Thank you for the response. I have no further questions at this time.

---

### Author Rebuttal · Authors · 2024-08-06

Thank you for your valuable and constructive feedbacks. We have performed the additional simulations as requested by the reviewers and have provided the results in this pdf file.

---

### Decision · Program_Chairs · 2024-09-25

**Decision:**

Accept (poster)

**Comment:**

The paper studies the problem of finding the core for the reward allocation problem when the information about reward functions is incomplete. Specifically, this paper assumes that we have access to an oracle that takes as input a coalition and outputs a stochastic reward for that coalition. The paper obtains an algorithm for the special case of strictly convex games which outputs a point in the expected core and with high probability uses at most a polynomial number of samples.

The paper studies an interesting problem, and provides an elegant solution for it. The paper is well-written. For this reason, we recommend
 acceptance. Anyway, it would be desirable that authors revise the paper by including more motivations and discussion about the large sample complexity, and the validity of the assumption of strict convexity, as highlighted by the discussion.